



# Pico-Light H₂O: Intercomparison of in situ water vapour measurements during the AsA 2022 campaign.

Mélanie Ghysels[1], Georges Durry[1], Nadir Amarouche[2], Dale Hurst[3,4], Emrys Hall[4], Kensy Xiong[3,4], Jean-Charles Dupont[5], Jean-Christophe Samake[2], Fabien Frérot[2], Raghed Bejjani[1], Emmanuel D. Riviere[1]

[1]Groupe de Spectrométrie Moléculaire et Atmosphérique (GSMA, CNRS UMR 7331), Université de Reims, UFR Sciences Exactes et Naturelles, Moulin de la Housse B.P. 1039, 51687 Reims Cedex 2,FRANCE
[2]INSU Division Technique, 1 place Aristide Briand, 92195 Meudon cedex, FRANCE
[3]Cooperative Institute for Research in Environmental Sciences, University of Colorado Boulder, United States
[4]NOAA GlobalMonitoring Laboratory, 325 Broadway R/GML1, Boulder, CO 80305, USA
[5]Site Instrumental de Recherche par Télédétection Atmosphérique (SIRTA), LMD - Ecole Polytechnique, Route de Saclay, 91128 Palaiseau Cedex, FRANCE

*Correspondence to*: Mélanie Ghysels (melanie.ghysels-dubois@univ-reims.fr)

**Abstract.** The mid-infrared lightweight stratospheric hygrometer, "Pico-Light H₂O", the successor of Pico-SDLA H₂O, is presented and its performances are evaluated during the AsA 2022 balloon-borne intercomparison campaign conducted at the CNES Aire-sur-l'Adour (AsA) balloon launch facility and the Aeroclub d'Aire-sur-l'Adour. The Pico-Light instrument has primarily been developed for sounding of the upper troposphere and stratosphere. Though, during the AsA 2022 campaign we expand the range of comparison including the lower troposphere additionally. Three different types of hygrometers and two models of radiosonde were flown, operated by the French Space Agency (CNES) and the NOAA Global Monitoring Laboratory (GML) scientific teams: Pico-Light H₂O, the NOAA Frost Point Hygrometer (FPH), the micro hygrometer (in an early phase of development), M20 and iMet-4 sondes. In this frame, we intend to validate measurements of Pico-Light H₂O through a first intercomparison with the NOAA FPH instrument. The in situ monitoring of water vapour in the UT-LS keeps being very challenging from an instrumental point of view because of the very small amounts of water vapour to be measured in these regions of the atmosphere. Between the lapse rate tropopause (11 to 12.3 km) and 20 km, the mean relative difference between water vapour mixing ratio measurements by Pico-Light H₂O and NOAA FPH was (4.2 ± 7.7) %, mean tropospheric difference is was (3.84 ± 23.64)%, with differences depending on the altitude range considered. In the troposphere, relative humidity over water (RH) comparisons leads to an agreement between both Pico-Light and NOAA FPH of -0.2% on average, with excursions of about 30% RH due to moisture variability. Expanding the comparison to meteorological sondes, the iMet-4 sondes agree well with both Pico-Light and FPH between ground and 7.5 km (within ± 3% RH) and so does for M20 sondes, up to 13 km, which are wet biased by 3% RH and dry biased by 20% in case of saturation.





## 1. Introduction

Water vapour plays an important role in the radiative balance on Earth since it is the principal source of infrared opacity. Its

contribution to the greenhouse effect is about 60 % to 75% (Kiehl and Trenberth, 1997; Schmidt et al., 2010; Lacis et al., 2013). Simulations based on radiative-convective models and observations have demonstrated that the surface warming caused by an increase in greenhouse gases like $CO_2$ could lead to a moistening of the troposphere (Dessler, 2013; Dessler et al., 2008; Dessler and Wong, 2009; Minschwaner and Dessler, 2004; Soden et al., 2005). This coupling could double the warming induced by $CO_2$ only (Banerjee et al., 2019; Dessler et al., 2013). Some models indicate that an increase in

tropospheric temperatures could increase stratospheric water vapour (SWV), implying the existence of a stratospheric water vapour feedback coefficient of about +0.3 $W.m^{-2}.K^{-1}$ (Dessler et al., 2013). Therefore, SWV has a great influence in the global radiative and chemical equilibrium. Various studies, based on radiative-chemical models, have shown a correlation between the variations in SWV and the changes in stratospheric ozone and the changes in stratospheric and mean global temperatures (Dvortsov and Solomon, 2001; Riese et al., 2012; Solomon et al., 2010). Stratospheric water vapour is a

significant contribution to the radiative equilibrium of the stratosphere, and therefore to the global radiative equilibrium. Observational studies have shown that a moistening of the stratosphere could lead to a warming of the mean surface temperature (Forster and Shine, 1999; Wang et al., 2017), with disparities at different latitudes.

In the upper troposphere and stratosphere, mixing ratios of water vapour are found between 2 and 5 ppmv. Hence, in this altitude range, the water vapour measurements are likely to be polluted by water vapour outgassing from the instrument or

from the balloon envelope. Altitudes higher than 15 km are only probed at high resolution by instruments carried under stratospheric balloons or selected number of high-altitude aircrafts. Satellite-borne observations, such as carried by Aura MLS, allows vertical profiling with a vertical resolution scaling from 1.5 to 3 km above 15 km with a high spatial and temporal coverage. Though the large coverage allows to address large-scale processes, to investigate regional or local processes, the use of in situ instruments remains the only way to proceed. So is the case for satellite validation activities.

Instruments such as the one described here, allows vertical resolution of a few meters, allowing to capture fine-scale signatures due to processes not well resolved in global models. Measuring such low abundance remains challenging. The GCOS (Global Carbon Observing System) requirements for the measurement of stratospheric profiles is a 5% uncertainty. Therefore, the differences between coincident measurements in this region have to agree within the 5% range. It is recognized that rigorous intercomparisons are of critical importance to allow a valuable scientific interpretation.


Comparisons between hygrometers have found discrepancies as large as 50% at the sub10 ppm level, well above the stated instrumental uncertainties (5 % -10%). In Vömel et al. (2007), the comparison between FLASH-B and the NOAA/CMDL hygrometers in the stratosphere lead to differences within ± 10% (0.5 ppmv) between 11 and 20 km, and as high as 30% for altitudes lower than 11 km and higher than 20 km. (Rollins et al., 2014) compares aircraft- and balloon-based in situ

hygrometers in the UTLS during the 2011 airborne intercomparison campaign MACPEX. Differences as large as 20% (0.8



ppmv) are found for mixing ratios below 5 ppmv, depending on the instrument pair considered and the volume mixing ratio probed. (Kaufmann et al., 2018) reported stratospheric comparisons of FISH and AIMS to an average reference value calculated from measurements from AIMS, FISH, SHARC and HAI. Measurements have been performed onboard the HALO aircraft. In the range below 10 ppmv, the mean differences between AIMS, FISH and the reference value falls within

the ± 15%. Deviations as large as 20% have been found in some cases, for which the reason remains unclear. In (Ghysels et al., 2016), we compared our former Pico-SDLA $H_2O$ hygrometer with the FLASH-B Lyman-α hygrometer in the tropical upper troposphere and stratosphere (15 to 23 km). The differences between both instruments was scaling from 0.5 to 1.9% (25 to 100 ppbv) above the cold point tropopause.

In this work, we report the development of a rugged, lightweight, laser diode hygrometer, "Pico-Light $H_2O$". Such a hygrometer launched on small balloons can make frequent measurements in difficult meteorological conditions and at moderate cost, thereby multiplying flight opportunities. The high resolution and accuracy of such instrument is a precious tool to investigate transport processes in the UTLS, where absolute modulation of the local mixing ratio scales are within 10 to 20% of the typical mixing ratio. Between 2019 and 2022, Pico-Light $H_2O$ was tested in seven flights from rubber balloons

at the Aire-sur-l'Adour (AsA) CNES balloon facility in south western France (43.7°N). In 2022, in the frame of the AsA 2022 campaign (https://www.hemera-h2020.eu/small-sensors-campaign/), the resulting measurements of water vapour were compared to the NOAA ESRL frost point hygrometer (NOAA FPH) from the ground up to few kilometres below the balloon burst altitude and to meteorological radiosondes (M20 and iMet-4) in the lower troposphere. Section 2 describes quickly Pico-SDLA $H_2O$, predecessor of the present Pico-Light $H_2O$ hygrometer. Section 3 describes in detail the Pico-Light $H_2O$

hygrometer, its performances and a discussion around instrumental uncertainties. Section 4 describes the NOAA FPH hygrometer. The M20 sondes are described in section 5. Sections 6 and 7 overview the datasets and flight conditions, then Sections 8 and 9 present the results of the intercomparisons, in volume mixing ratio between Pico-Light $H_2O$ and NOAA FPH (section 8) and in relative humidity in the troposphere between the in situ hygrometers and meteorological sonde (section 9).

**2. Predecessor instrument, Pico-SDLA $H_2O$**

Pico-Light $H_2O$ is the lightweight successor of Pico-SDLA $H_2O$ (Durry et al., 2008), which we will briefly describe. This balloon-borne spectrometer was tested against other hygrometers, both in flight and in an atmospheric simulation chamber (Behera Abhinna K. et al., 2018; Berthet et al., 2013; Durry et al., 2008; Fahey et al., 2014; Ghysels et al., 2016; Korotcenkov, 2018). With the Pico-SDLA, the beam of a 2.63 µm antimonide laser diode, was propagated in the open

atmosphere over a 1-m distance; absorption spectra were thereby recorded in situ at 1s - intervals. The water vapour mixing ratio were retrieved from the in situ absorption spectra using a molecular model in conjunction with in-situ atmospheric pressure and temperature measurements. The laser wavelength was chosen by adjusting the driving current and the temperature of the laser semiconductor (by means of a Peltier thermoelement). The ramping of the driving current allowed





the scanning of the laser wavelength over the full molecular line shape of the selected $H_2O$ molecular transition (Durry and Megie, 2000). With a weight around 8.5kg, Pico-SDLA $H_2O$ was operated from medium-size balloons or as a piggy-back onboard large scientific gondolas. Table 1 compares Pico-SDLA $H_2O$ to the new Pico-Light $H_2O$.

**Table 1: Comparison of Pico-Light H₂O with its predecessor Pico-SDLA H₂O.**

|  | Pico-SDLA H₂O | Pico-Light H₂O |
|---|---|---|
| Total mass (kg) | 8.5 | 2.7 |
| Electronics mass (kg) | 2.5 | 1.4 |
| Optical cell mass (kg) | 3.5 | 0.7 |
| Data points per spectrum | 256 | 1024 |
| Energy Consumption (Wh) | 9 | 3.5 |
| Laser wavelength (μm) | 2.63 | 2.63 |
| Optical length in ambient air (m) | 1.00 | 1.00 |

The novelty of Pico-Light $H_2O$ lies in its new electronics, lighter weight (2.7 kg), simpler mechanical structure, and improved energy management. The dramatic weight reduction made it possible to fly the instrument from a small rubber weather balloon. The flight duration under such a balloon is about 2 hours, while that of a medium-size balloon is 4 to 8 hours. Thus, the instrument is subjected to low temperatures (down to – 70°C at the tropopause) during a shorter period of time, compared to the Pico-SDLA instrument.

## 3. Pico-Light H₂O

The development of the Pico-Light $H_2O$ hygrometer began in 2017 with the support of CNES and CNRS. Figure 1 shows the hygrometer and its launch under a 1200 g Totex rubber balloon from the CNES Aire-sur-l'Adour facility (France). The hygrometer was launched twice in 2019 and five times in 2022. The flights occurred within the HEMERA Work Package 11 (WP11). Pico-Light $H_2O$ has primarily been developed for frequent soundings of the UTLS, relying on the optimization of the optical cell design (minimizing contamination) and absorption line selection (line intensity and width).





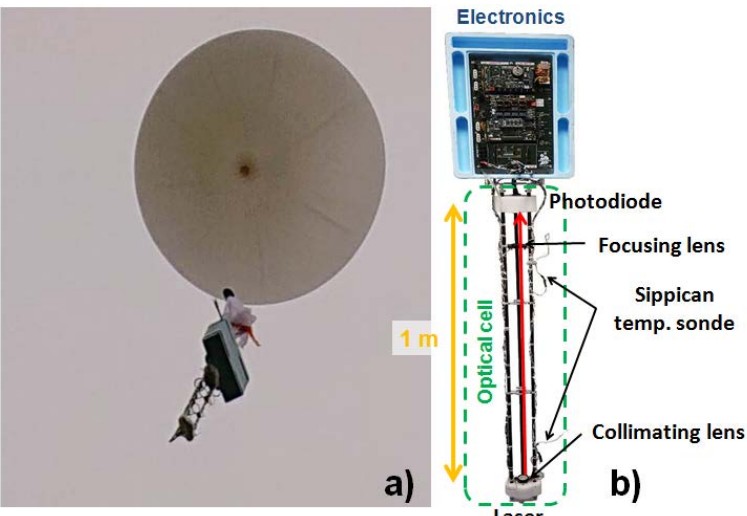

**Figure 1: Pico-Light H$_2$O during its launch under a 1200g rubber balloon from the CNES Aire-sur-l'Adour facility in 2019 and 2022 (France). (b) Detailed picture of the hygrometer.**


The Pico-Light H$_2$O hygrometer offers several advantages compared to closed-cell spectroscopic techniques. The fast acquisition and short response time (typically a few milliseconds) allow the monitoring of fast humidity changes since there is no need for equilibrium with any cell surfaces. The open optical path reduces contamination of the measurements by water desorbed from the walls of a closed cell. Moreover, the used technique is highly selective as a specific water vapour

rovibronic transition is swept over by the laser, which is free from overlapping by other molecular species. Like all optical absorption hygrometers, the instrument is highly sensitive, even at low mixing ratios. Finally, the large dynamic range provides the mixing ratio profile from the ground to the burst altitude, at the difference of other techniques.

### 3.1. Electronics

The enhanced electronics utilized in the current version are both smaller and more energy-efficient compared to those employed in Pico-SDLA. This decrease in power consumption and shorter flight duration have contributed to a significant one-third reduction in the energy budget, now standing at just 3.5 Wh. As a result of these reduced power requirements, we have managed to cut down the battery weight by one sixth, resulting in a mere 0.4 kg.). The shorter flight also meant that the electronics box needs less thermal insulation, and its heat capacity is sufficient to keep its temperature above the minimum

operating temperature during the entire flight. Using less thermal insulation contributed to the decrease in weight. Figure 2 shows the instrument architecture.





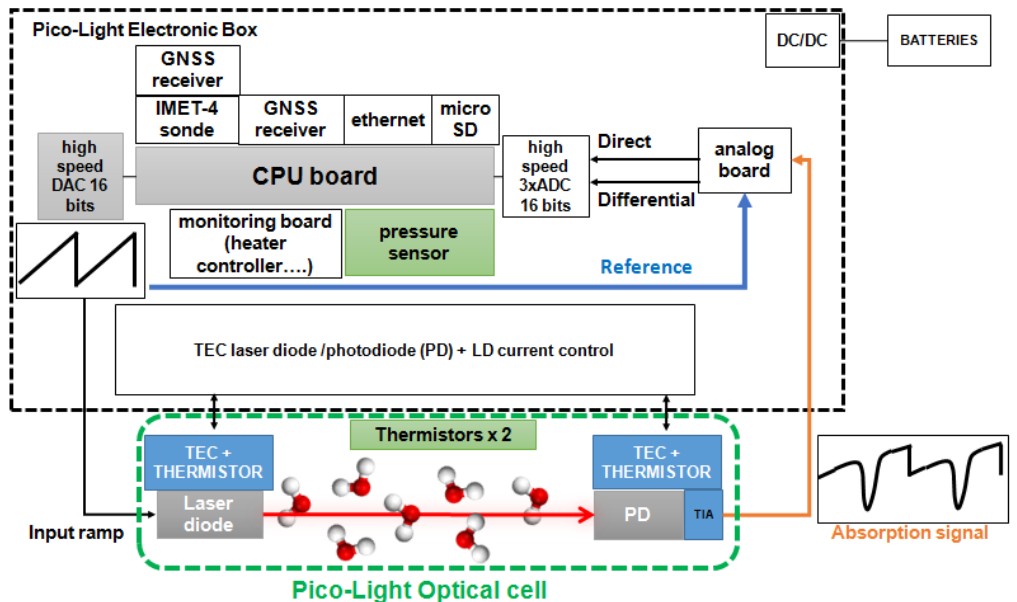

**Figure 2: Schematic of the Pico-Light H$_2$O electronics architecture.**


The electronics box includes an environmental sensor (InterMet model iMet-4). Its temperature and pressure readings are not used, but its humidity measurements provide a useful check on the laser hygrometer's operation in the lower troposphere. The Pico-Light electronics also include a GNSS (Global Navigation Satellite System) that tracks the instrument position throughout the flight. A second GNSS that is included in the environmental sensor can be used if the first one fails. The

ambient pressure is measured by an absolute pressure transducer (Honeywell PPT1; precision 0.05% full scale, absolute uncertainty 0.5 hPa). The retrievals are obtained using the Honeywell pressure as an input in the spectrum processing. The pressure measurements allow the software to detect whether the balloon is ascending or descending. Measured parameters such as pressure are used in various control loops. If the value of one of the parameters is anomalous, the software turns off the associated loop and sets the parameter back to its initial value.


The acquisition module controls the gains of the analog signal chain and the temperatures of both the detector and the diode laser. It also generates the laser current ramp. One measurement consists of three signals that are recorded and digitized simultaneously using a 16-bit ADC on a parallel bus interfaced with a microcontroller: (1) the direct atmospheric signal, (2) the ramp signal, and (3) the differential signal. The direct atmospheric signal is the absorption spectrum recorded by the

detector, which is a cooled InAS photodiode (Judson Technologies Inc.). The ramp signal scans the laser wavelength by modulating its driving current. The ramp is generated by a 16-bit high speed DAC controlled by an SPI bus. Unfortunately, ramping the laser driving current also ramps the power of the laser, resulting in a sloping background in the atmospheric spectra. Hence, we record also the analog difference between the atmospheric signal and the ramp signal (Durry and Megie,

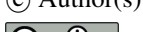

2000; Durry et al., 2000). The relative contribution of the two signals is balanced so as to minimize the sloping background. The differential signal is used to determine the position of the absorption line peak during the flight; any spectral drift is compensated by adjusting the temperature of the laser diode. It has been stressed that only the direct atmospheric signal is used for mixing ratio retrievals.

During the flight, the embedded software operates the instrument with no human control, and no telecommand/telemetry is necessary. The software includes all parameters necessary for the automatic tasks, e.g. the frequency detuning of spectra according to the ambient pressure and the selection of the absorption line according to the altitude. During ascent and descent, the software adjusts the amplitude of the diode current modulation and selects the proper water vapour absorption line based on the ambient pressure. The proper diode temperatures, corresponding to each of the two lines, are set onboard depending on the measured ambient pressure. The diode laser temperature is stabilized to within 10 mK over one scan. Therefore, the software knows which of the two lines has to be used at one given pressure to optimize the measurement uncertainty. Essential diagnostics and information about the physical environment are stored on a micro-SD drive along with the data.

### 3.2. Optical cell

The optical cell, shown in Fig. 1, includes the 1-m open structure beneath the electronic enclosure, which allows the laser beam to measure the ambient air parcel directly with no interference from water outgassed by the electronics. The mechanical design of Pico-Light $H_2O$ differs significantly from that of its predecessor Pico-SDLA $H_2O$. The mechanical structure of Pico-SDLA $H_2O$ comprised six carbon fibre composite tubes, with three large braces made of PVC. The structure of Pico-light $H_2O$ comprises only three carbon fibre composite tubes, with three small braces made of aluminium. The high strength-to-weight ratio of the tubes strengthened the instrument against the mechanical stress of a flight and collision damage when landing. These modifications lightened the optical cell by a factor of five, from 3.5 kg to 0.7 kg.

The ambient temperature is measured using two fast-response temperature sensors (Sippican) with an uncertainty of 0.2°C rms and a resolution of 0.1 °C. Their uncertainty was improved by an intercomparison program with WMO (Organization (WMO) et al., 2011). One sensor is located at each end of the optical cell. Each temperature measurement is the average of 20 readings made during 1 ms, with outliers removed. The time between measurements is sufficiently short so that successive measurements during a flight differ by less than 0.05°C.

Despite having an innovative reflective coating, the temperature sensors were susceptible to solar and infrared radiation even though the sun elevation during the descents of the balloon was between 11 and 20 degrees. Therefore, corrections for radiation heating were applied according to the manufacturer's datasheet based on WMO intercomparison's results. The temperature used to analyse the spectra was the lower of the two values because the corresponding sensor was assumed to be less affected by solar radiation.





The collimating and focusing lenses are heated to avoid the formation of ice or dew.

### 3.3. Spectra acquisition and laboratory testing

The $H_2O$ measurements are taken at 1 s intervals. During that interval, 200 ms are devoted to record the elementary
atmospheric spectrum (within this time frame, 5 spectra are recorded), which comprises 1024 data points. The remaining 800
ms are used to record the atmospheric pressure and temperature, the GPS data, and the status of the instrument (internal
temperatures, electronics gains, laser current and temperature, etc.).

The average descent speed varies from close to 35 m/s at the ceiling altitude, down to 5 m/s in the lower troposphere. Then,
the vertical resolution of the measurements varies accordingly from 35 m to 5 m.

During a flight, the temperature gradient encountered during the dynamic phase of the flight is able to cause a drift of the
laser wavelength, leading to the acquisition of unusable spectra. To overcome this issue, the laser diode is mounted into a
thermally-insulated enclosure, equipped with a heater and a thermistor. This temperature-controlled enclosure is able to
stabilize the laser wavelength despite the severe temperature range encountered during a flight (from +30°C at ground level
to -70°C at the tropopause). The photodiode enclosure has a similar temperature control. The appropriate laser current and
temperature are determined from both room temperature and cold environment calibration in an environmental chamber.
During a flight, the onboard software uses the measured atmospheric temperature to further stabilize the temperatures of both
the photodiode and the laser, achieving a stability better than 5 mK.

### 3.4. Spectroscopy

The mixing ratio is extracted from the atmospheric absorption spectrum using a non-linear least-squares fitting algorithm
applied to the full line shape, based on the Beer-Lambert law and in conjunction with in situ pressure and temperature
measurements (Durry and Megie, 1999). The molecular line shape is modelled using a Voigt profile (VP). Fitting the VP to
the measured spectrum yielded residuals consistent with the instrument noise. No systematic residuals caused by higher-
order line shape effects were observed at stratospheric pressures (our main region of interest) or at higher pressures, in the
middle troposphere. For a water vapour line broadened by air, collision-induced fine effects can cause the VP to be
inadequate in some spectral regions. These effects are observed for other molecular species also. Some of these effects have
been reported in spectroscopic studies (Delahaye et al., 2019; Devi et al., 2007b, a; Galatry, 1961; Ghysels et al., 2013, 2014;
Hartmann et al., 2009; Joubert et al., 2002; Lamouroux et al., 2015; Lance et al., 1997; Lisak et al., 2003, 2015). The impact
of such inadequacy on the uncertainty budget depends on the signal-to-noise ratio (SNR) of the spectra. An advanced line
profile will improve the retrievals if the SNR is sufficient to extract the additional line parameters. In the present case, for the
lines considered, and due to the spectra SNR, such high order effects are not noticeable in our region of interest, and thereby
the induced bias remains impossible to estimate. However, the line area is the predominant factor in the determination of the
mixing ratio since the full line shape is used in the fit. The two water vapour lines used here are isolated (not affected by
line-mixing (Hartmann et al., 1996)) and have limited interference with neighbouring lines. Possible non-Voigt effects





include Dicke narrowing, having its largest influence at low pressures, and speed dependence of the collisional width, having
its largest influence at high pressures. The uncertainty related to these high-order effects ranges at the sub-percent level
(~0.1%). Neither effect is discernible in the residuals, unlike for our carbon dioxide and methane sensors, where more
advanced line shapes are used together with spectroscopic parameters previously determined in laboratory.

From the ground to the balloon burst altitude, the water vapour mixing ratio varies by several orders of magnitude, from
about 4 ppmv in the stratosphere to several thousand ppmv at ground level. Therefore, two spectroscopic transitions are
needed to probe both troposphere and stratosphere: the $2_{02} \leftarrow 1_{01}$ and the $4_{13} \leftarrow 4_{14}$ lines, each of them suitable for a given
range of concentration. The selected rotation-vibration $H_2O$ transitions are the same as for the former Pico-SDLA
instrument. For measurements from the ground to around 260 hPa, the $4_{13} \leftarrow 4_{14}$ $H_2^{16}O$ line at 3802.96561 $cm^{-1}$ is used.
Above the 260 hPa level, the $2_{02} \leftarrow 1_{01}$ $H_2^{16}O$ line at 3801.41863 $cm^{-1}$ is used. Both sets of line parameters are obtained from a
laboratory study (Durry et al., 2008). The uncertainty of each line intensity is 0.85 % and 0.95% respectively for the
$2_{02} \leftarrow 1_{01}$ $H_2^{16}O$ line and the $4_{13} \leftarrow 4_{14}$ line.

The spectra processing is divided into four steps: (1) the search for off-line data points used for baseline interpolation, (2)
construction of the absolute frequency axis of the spectra, (3) determination of a first approximation of the mixing ratio
based on peak absorbance, (4) non-linear least square fitting of atmospheric spectra. The selection of the off-line data points
is realized by calculating the first derivative of the signal. The minimum and maximum of the derivative gives the positions
of the points at the line half maximum. The points used for baseline interpolation are those located away from the line center
by 3 times the full width at half maximum. In the stratosphere, the probed line is well isolated. At pressures higher than 350
hPa, the wings of the $2_{02} \leftarrow 1_{01}$ $H_2^{16}O$ line have a significant contribution to the measured transmittance of the $4_{13} \leftarrow 4_{14}$ $H_2^{16}O$
line. This contribution is therefore included in the fitting procedure. The transmittance of these lines is calculated based on
the measured pressure and temperature and the retrieved mixing ratio. In the considered spectral range, the contribution of
other greenhouse gases is negligible since line intensities are below $10^{-23}$ $cm^{-1}/molecules.cm^{-2}$, which is 2 to 3 orders of
magnitude smaller than the line intensities of the probe lines. A 4th-order polynomial function is then used to interpolate the
spectrum baseline on a first approximation. Dividing the atmospheric spectrum by this polynomial baseline allows one to
calculate an approximate transmittance and to estimate a first approximation of the mixing ratio (using the peak absorbance
value), which will be used as an input of the non-linear least square fitting procedure. The absolute frequency axis of the
spectrum is calculated using spectroscopy: the spectrum peak position coincides with the absolute line center frequency
found in the HITRAN spectroscopic database (Gordon et al., 2017). Knowing the ambient pressure and temperature and
using the first approximation of the volume mixing ratio allows one to simulate a synthetic spectrum from which the
absolute frequencies of the points at the half maximum are known. The frequency step between each data point is then equal
to the absolute frequency width at half maximum (obtained from the synthetic spectrum) divided by the number of
experimental points at half maximum (estimated from the approximate experimental transmittance, as describe above). The
absolute frequency step is assumed to be constant over the full scan.





During the spectra processing, the standard deviation of the fitting residuals is calculated. This acts as a criterion of the
quality of the fit. Only the retrievals associated with a standard deviation within the measurement noise are conserved.

### 3.5. Uncertainties

Table 2 lists the vertical resolution and the measurement uncertainties of Pico-Light $H_2O$ per levels from the ground up to
the balloon burst altitude.

The noise of one spectrum is about $5\times10^{-4}$ in absorption units. The corresponding signal-to-noise ratio in the stratosphere is
about 2000. The uncertainty is calculated at each pressure from the standard deviation of the fit at a given pressure level.
Using unitary spectra (no averaging), the standard deviation of the mixing ratio in the stratosphere, and therefore, the
precision, is about ±277 ppbv. For a one-second averaging time, the precision is ±130 ppbv (co-addition of 5 spectra). By
comparison, frost control instabilities of the NOAA FPH dominate the measurement uncertainty budget (±2σ), ranging from
±10% in the lower troposphere to ±2% (±100 ppbv) in the stratosphere (Hall et al., 2016).


Random uncertainties are dominated by spectrum deformations due to mechanical vibrations, especially at high altitudes,
where the absorbance of the probing line is the smallest. To limit this effect, the simple and robust design of the optical cell
minimizes mechanical vibrations during the flight, thereby limiting strong variations of the spectra baseline. Still, the more
severe deformations are observed after the balloon bursts, when the vertical speed is the fastest and the mechanical
vibrations of the instrument structure are the strongest. In this range, the random uncertainty can reach up to ±10%, leading
to larger dispersion of the measurements seen down to few kilometers below the balloon burst altitude. It quickly decreases
as the balloon continues to descent.

A second area of significant distorsions is found where winds are strong above the hygropause. In this range, the volume
mixing ratio is within its lowest values and mechanical vibrations are getting important. Wind speed is found above 17m/s.
In this case, the random uncertainty is on average 3.5%, the largest uncertainty being found at the hygropause (3.8% in Table
2). Below the hygropause, while mixing ratio is increasing, the distorsions induced in case of strong winds see their
influence quickly reducing. In any case, the random uncertainty induced by the baseline variability is variable, its influence
depending on the line absorbance and the severity of the baseline deformation.

Systematic errors are dominated by uncertainties in the estimation of the peak absorbance of the spectrum as well as the line
intensities (~1%). Systematic errors due to the error in the peak absorbance dominates the budget down to 120 hPa (15.3
km). Two absorption lines are used for the measurements, as stated in the section 3.4. For pressure levels higher than 260
hPa (upper troposphere and stratosphere, altitudes higher than 10 km), for a characteristic mixing ratio of about 5 ppmv, at
50 hPa, the absorption depth is of about 0.013. The subsequent systematic uncertainty is of about 3.9%. At 260 hPa (about
10 km), where mixing ratio rises up to few hundreds of ppmv, the absorption depth is of about 0.46. The subsequent





systematic uncertainty is of reduced to 0.1%. For altitudes below 10 km, a second absorption line is used instead, having a line strength 10 times smaller than the "stratospheric" line. This is to compensate for the dramatic increase of water vapour mixing ratio. In the lower troposphere, where mixing ratios quickly rise up to several hundreds to thousands of ppmv, the molecular absorption is always larger than 0.15. At 7 km (pressure level : 430 hPa), the absorption depth of the selected line

is of about 0.20. The subsequent systematic uncertainty is of about 0.3%. Decreasing in altitude, where mixing ratios dramatically increase, the systematic uncertainty decrease. From about 15 km, the systematic uncertainty due to spectroscopy starts to dominate the budget and is kept as a constant uncertainty.

Biases in the environment's pressure and temperature measurements generate additional minor systematic errors.

The maximum uncertainties of the pressure and temperature measurements are of 0.5 hPa and 0.2K respectively. Pressure measurements are obtained by averaging the instantaneous pressure measurements, from the Honeywell PPT1 sensor onboard, over half a second. This process allows to improve the pressure measurement's uncertainty. The subsequent systematic uncertainty on the retrieved mixing ratio scales from 0.03 % at ground level, to up to 0.3% at the float altitude. The influence of pressure on the systematic uncertainty decreases quickly between the burst altitude and about 100 hPa

(about 16.5 km). Taking into account errors in the estimation of the baseline and the frequency axis, the global systematic uncertainty increases from 0.4% at ground level up to as much as 0.7% on average in the middle stratosphere.

The total systematic uncertainty is obtained by summing all the above-cited uncertainties in quadrature. The systematic uncertainty varies from 1.0 % at ground level (fixed to line intensity uncertainty) up to 6.3% at 45 hPa and above. Random

uncertainty varies from 0.02% at ground, up to 10% in the stratosphere.

**Table 2. Systematic ($u_S$) and random ($u_r$) uncertainties on mixing ratio X made by Pico-Light $H_2O$. Also shown is the resolution $\delta(z)$ of the height z. The "hygr.." flag indicates the altitude range of the hygropause in this case.**

| P [hPa] | δ(z)[m] | $u_S(X)$[%] | $u_r(X)$[%] | Total uncertainty [%] |
|---|---|---|---|---|
| 10-45 | 25 | 6.3 | 10.2 | 12.0 |
| 46 - 69 | 20 | 4.1 | 6.3 | 7.5 |
| 70- 83 | 15 | 4.0 | 3.3 | 5.2 |
| 84 - 100 | 13 | 3.3 | 1.0 | 3.5 |
| 100 -120 | 12 | 2.7 | 3.6 | 4.5 |
| 121-150 | 8 | 2.2 | 3.8 - hygr. | 4.4 |
| 150-180 | 8 | 1.5 | 3.0 | 3.4 |
| 180- 350 | 7 | 1.0 | 1.3 | 1.6 |
| 351 - 620 | 5 | 1.0 | 0.8 | 1.3 |
| 621-1000 | 3 | 1.0 | 0.7 | 1.2 |



## 4. The NOAA frostpoint hygrometer

The NOAA FPH (see Fig. 3) is a balloon-borne instrument that makes in situ measurements of atmospheric water vapour vertical profiles up to altitudes of ~ 28 km. The basic measurement principle and calibration method have remained unchanged since 1980, although the instruments have been significantly modernized over the years (Hall et al., 2016). The chilled mirror principle relies on creating and maintaining a thin, stable layer of condensed-phase water (dew or frost) on a highly reflective mirror through rapid feedback temperature control of the mirror. Constant cooling of the mirror is provided by a copper cold finger immersed in liquid cryogen (R23). At the other end of the cold finger is the chilled mirror that extends into the path of air flowing at 3–6 m s−1 through the instrument. Intermittent heating of the mirror is provided by an electrified nichrome wire wrapped around the narrow shaft of the mirror. An infrared LED and photodiode serve as the optical source and detector, respectively, for rapid measurements of the mirror's reflectivity that is affected by the amount of condensate on its surface. When too little (much) light is reflected, the amount of condensate on the mirror is reduced (increased) by heating (cooling) it. A calibrated thermistor embedded in the mirror accurately measures the frost (dew) point temperature when the condensate layer is stable, indicative of equilibrium between the mirror's condensate layer and the water vapour in the air flowing over it. The partial pressure of water vapour in the flowing air is determined from the frost (dew) point temperature, then divided by the ambient atmospheric pressure to calculate the water vapour volume mixing ratio. Conceptually, frost point hygrometry allows water vapour mixing ratios to be determined from high-accuracy temperature measurements, eliminating the need for water vapour calibration scales or gas standards, which are notoriously difficult to maintain over decadal timescales (Hurst et al., 2011, 2016).

Balloon-borne FPH soundings over Boulder during the last 42 years have produced the longest continuous record of stratospheric water vapor (SWV) in the world. This "Boulder record", based on FPH data from 557 individual soundings, indicates considerable inter-annual variability in SWV and a net increase of ~0.8 ppmv (20%) since 1980. Profiles of SWV from NOAA FPH soundings at three sites – Boulder, Colorado, Hilo, Hawaii, and Lauder, New Zealand – have been routinely compared to those produced by the Microwave Limb Sounder (MLS) aboard the *Aura* satellite. The MLS has provided daily, near-global (82°S-82°N) measurements of SWV and other trace gases since August 2004. After 2010, the comparisons at all three sites began to show divergences between the FPH and MLS version 4.2 (v4) data sets that were increasing with time. In 2021, the MLS science team produced a new version 5.1 (v5) data set with the intention of reducing the positive (wet) MLS biases relative to the FPH. In all cases, the transition from MLS v4 to v5 retrievals resulted in MLS now having a negative (dry) bias relative to the FPH.



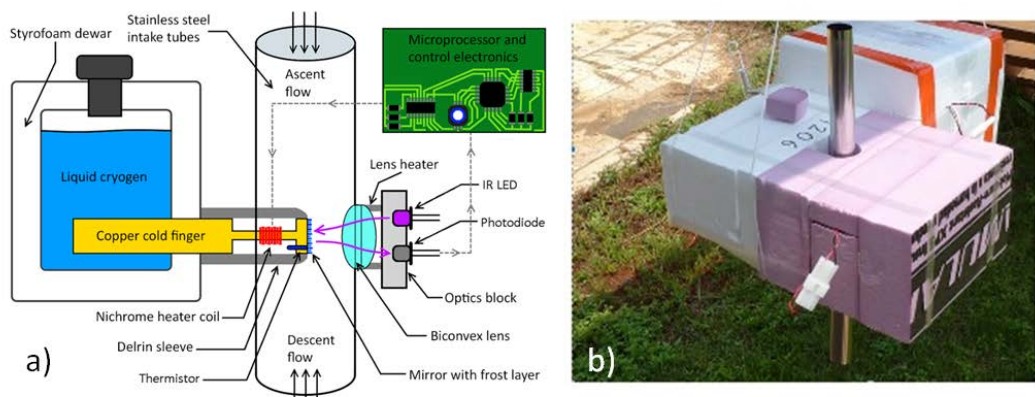


**Figure 3: Schematic of the NOAA FPH (Fig. 1 of (Hall et al., 2016)). b) Picture of NOAA FPH with an ozone sonde.**

## 5. M20 sondes from MeteoModem

The M20 is one of the radiosondes developed and sold by Meteomodem (Ury, France). The M20 was released in 2021, and it is fully compatible with the SR10 receiver system and the Meteomodem software. Its dimensions are 98mm x 63mm x 42

mm, weighing just 36 g with a lithium battery. The M20 is composed of a capacitive humidity sensor covered by an innovative metal coated shield that allows good ventilation while protecting the sensor from direct radiation and freezing water droplets. A temperature sensor measures the air temperature and is positioned at the very end of the sensor boom. A GNSS system provides measurements of the position, from which the pressure, the vertical velocity, the wind speed and direction are derived. The capacitive humidity sensor is composed of three primary components: a basic layer that acts as an

electrode; a dielectric material, whose characteristics are a function of relative humidity; and a fast response porous electrode that acts as the second electrode of the capacitor. A second thermistor is located under the protective shield close to the humidity sensor in order to have an approximative measurement of the temperature of the capacitive humidity sensor. The UPSI France Company is the subcontractor for this capacitive humidity sensor and these sensors are made specifically and exclusively for Meteomodem. Meteomodem radiosonde technology is used in 28 countries. All Meteomodem stations

installed since 2011 use the M10 and M20 technology. Before 2011, sites used the former version of M10 (M2K2), and by now all are using the M10/M20 technology. Most of these sites produce two M10 or M20 radiosoundings per day (Dupont et al., 2020).

## 6. Descriptions of the compared datasets

The water vapour datasets originate from three different types of instruments: an optical hygrometer (Pico-Light H$_2$O) and a

frost point hygrometer (NOAA FPH), with both able to provide reliable measurements of water vapour up to the middle



stratosphere, and meteorological radiosondes sondes from Intermet (iMet-4) and Meteomodem (M20) that provide accurate RH measurements up to about 6 to 13 km respectively.

Flight chains schematics are shown in Fig. 4. The Pico-Light hygrometer is able to provide reliable measurements during the balloon ascent and the descent under parachute. The flight chain separates the instrument from the balloon by 17 m. In this case, the ascent measurements become affected by outgassing from the balloon envelope at about 14 km. Below, measurements from the ascent and the descent of Pico-Light agree to within 3 %. These differences observed between the ascent and the descent arise from the rapid, natural changes in water vapour that are observed in the troposphere. In the stratosphere, the variability in water vapour is dramatically reduced, so the ascent and descent stratospheric vertical profiles are expected to be identical except the ascent profile is often contaminated by water vapour outgassing from the balloon envelope. Nevertheless, the descent measurements allow to reliably probe water vapour continuously from the lower troposphere up to few km below the balloon burst altitude. The NOAA FPH served as the reference instrument for our study, providing measurements during both ascent and descent. Indeed, the flight chain "unreeler" used puts a distance of about 36 m between the balloon and the instrument, dramatically reducing the influence of balloon outgassing to contaminate the ascent measurements up to an altitude of about 26 km. The FPH flight chain normally includes a pressure-activated valve that releases helium from the balloon in the middle stratosphere, allowing the instrument and unburst balloon to slowly descend and make contamination free measurements (Hall et al., 2016). This valve was used only for the FPH flight of September 19[th] (Table 3) because it greatly increased the distance of balloon travel from Aire-sur-l'Adour (Fig. 5), making payload recovery more difficult.

Radiosonde pressure, temperature, and horizontal wind (from GPS) measurements are accurate up to balloon burst (> 30 km), but the relative humidity measurements often lack the precision and accuracy necessary for stratospheric research and are therefore most useful in the low to mid-troposphere. Onboard Pico-Light, an iMet-4 meteorological sonde provides measurements of the instrument's position during the ascent and the descent, as a backup in case of failure in the GNSS system. On the other hand, the M20 radiosondes, launched under their own balloons, provides measurements during the ascent and descent.





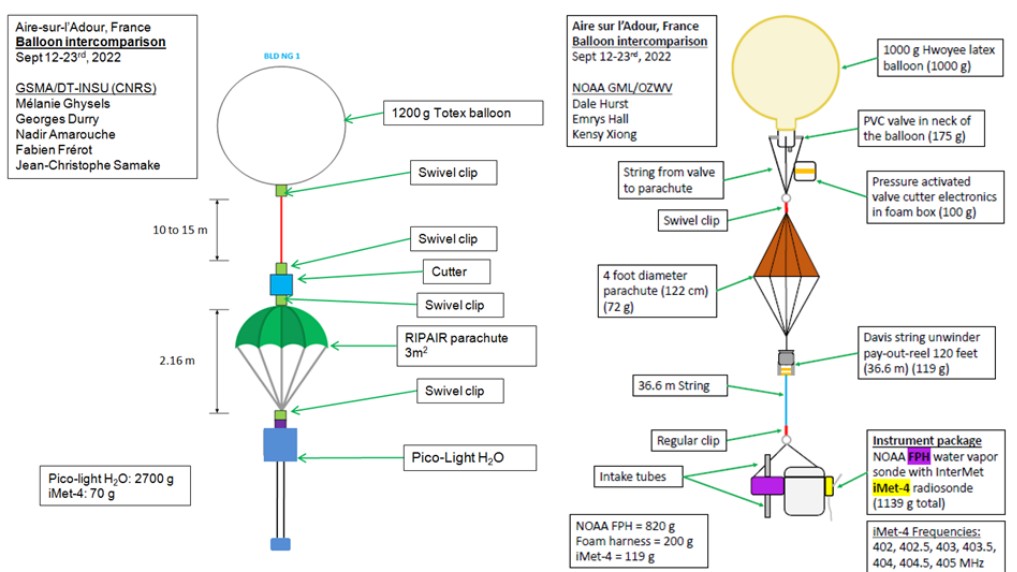


**Figure 4: Flight chains carrying Pico-Light H₂O (left) and the NOAA FPH (right) hygrometers.**

## 7. Flight conditions

The campaign occurred from September 12 to 23, 2022 from the CNES Aire-sur-l'Adour facility and from the Aéroclub d'Aire-sur-l'Adour in Southern France. The NOAA FPH has been launched from the Aéroclub launch pad and the Pico-Light

hygrometer (with iMet-4 attached) and M20sondes were launched from the CNES launch facility, 600 m away. In total, 15 flights have been realized, spread over 4 sessions of 3 to 4 balloons launched within 30 minutes. In this study, we focus on 3 sessions, which have occurred on September 19, 21 and 23, 2022. At these dates, both Pico-Light H₂O and NOAA FPH have been launched, therefore an intercomparison is possible.

**Table 3: Dates of flights together with the flights characteristics.**

| Date of flight | Instrument | Burst altitude (km) | Slow descent | Lapse rate tropopause altitude (km) (LRT) | Cold point tropopause altitude (km) (CPT) |
|---|---|---|---|---|---|
| | Pico-Light H₂O | 24.8 | N | | |
| 19/09/2022 | NOAA FPH | 27.7 | Y | 11.51 | 16.48 |
| | M20 sonde | 33.3 | N | | |





| | Pico-Light H$_2$O | 30.2 | N | | |
|---|---|---|---|---|---|
| **21/09/2022** | NOAA FPH | 25.8 | N | 11.07 | 16.35 |
| | M20 sonde | 36.9 | N | | |
| | | | | | |
| | Pico-Light H$_2$O #1 | 29.5 | N | | |
| | Pico-Light H$_2$O #2 | 30.2 | N | | |
| **23/09/2022** | NOAA FPH | 26.8 | N | 12.29 | 13.13 |
| | M20 sonde | 35.8 | N | | |

Table 3 lists the balloon burst altitude and the descent conditions for each flight. On the 21[st] and 23[rd], due to strong winds in the stratosphere, a slow (valved) descent was not permissible for the NOAA FPH balloons. For each landing of Pico-Light H$_2$O and NOAA FPH, we found no damage to the recovered instruments, therefore they were flown again on subsequent flight. However, if the Pico-Light H$_2$O or FPH instruments were to land in water, this would require replacement of the laser diode module and photodiode (Pico) and frost control electronics (FPH). For dry landings, the instruments are budget-friendly because, given the rugged optical and electronic components, the only the mechanical structures would need repair.

The trajectories from each instrument, for the flights used here for intercomparison (i.e. the 19[th], 21[st] and 23[rd] of September), are shown in Fig. 5. For the flights on September 21 and 23, both Pico-Light and NOAA FPH balloon trajectories remained within the same area, and the distance between the balloons was less than 15 km. On September 19[th], the slow (valved) descent of the NOAA balloon brought the instrument to land near Toulouse, 50 km away.

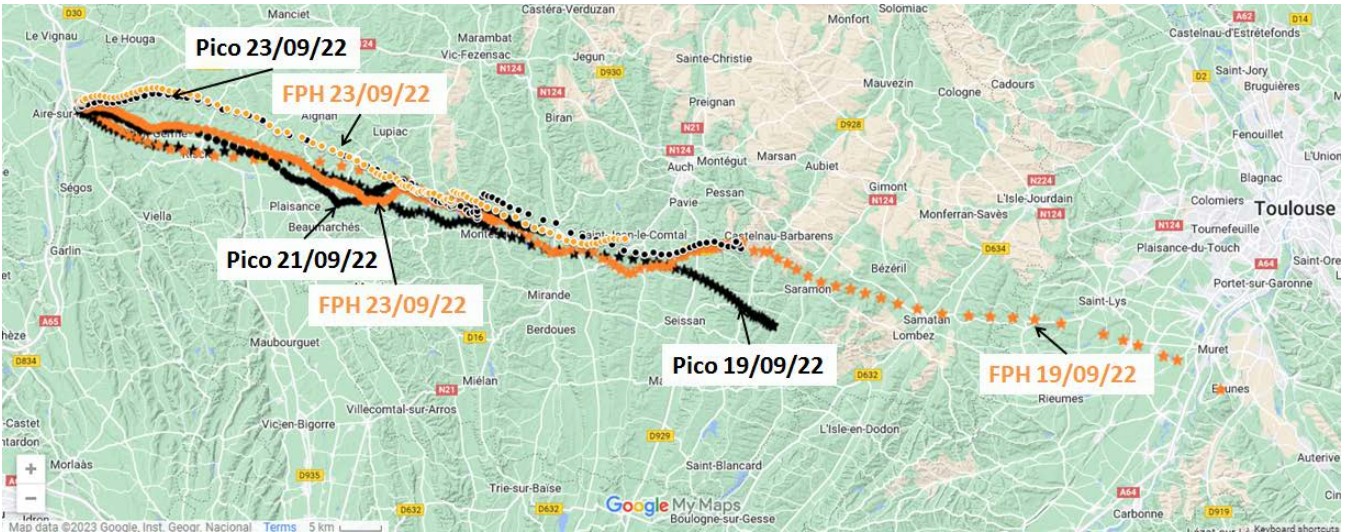

**Figure 5: Trajectories of balloons carrying Pico-Light H$_2$O and NOAA FPH during the campaign.**





## 8. Comparison between Pico-Light H₂O and the NOAA FPH

From September 19th to 23rd, upper tropospheric and stratospheric conditions above south western France changed quickly. During September 19th and 21st, a double tropopause structure was observed with a lapse rate tropopause height of ~ 11.5 km and a cold point tropopause height of 16.4 km. Between the 21st and the 23rd, the tropopause structure changed to a

single level with the lapse rate tropopause at 12.3 km and the cold point tropopause at 13.1 km.

In general, for all flights, structures on vertical profiles are observed by both the Pico-Light H₂O and the NOAA FPH water vapour measurements. Figure 6 is an illustration of the good agreement between the two instruments although contamination from outgassing is observed from 18 to 21 km on Pico-Light measurements. A discussion around the outgassing contamination is found later in the manuscript. A moist layer of 5.9 ppmv is observed by both instruments

between 14 and 17 km on the 21st, whereas on the 19th, the mean mixing ratio varies from 4.5 to 5 ppmv. An exception is found at 15.6 km where a thin 500 m-thick hydrated layer is also visible on the 19th, again by both instruments at the same altitude, which is not seen in ERA 5 (Fig. 7).

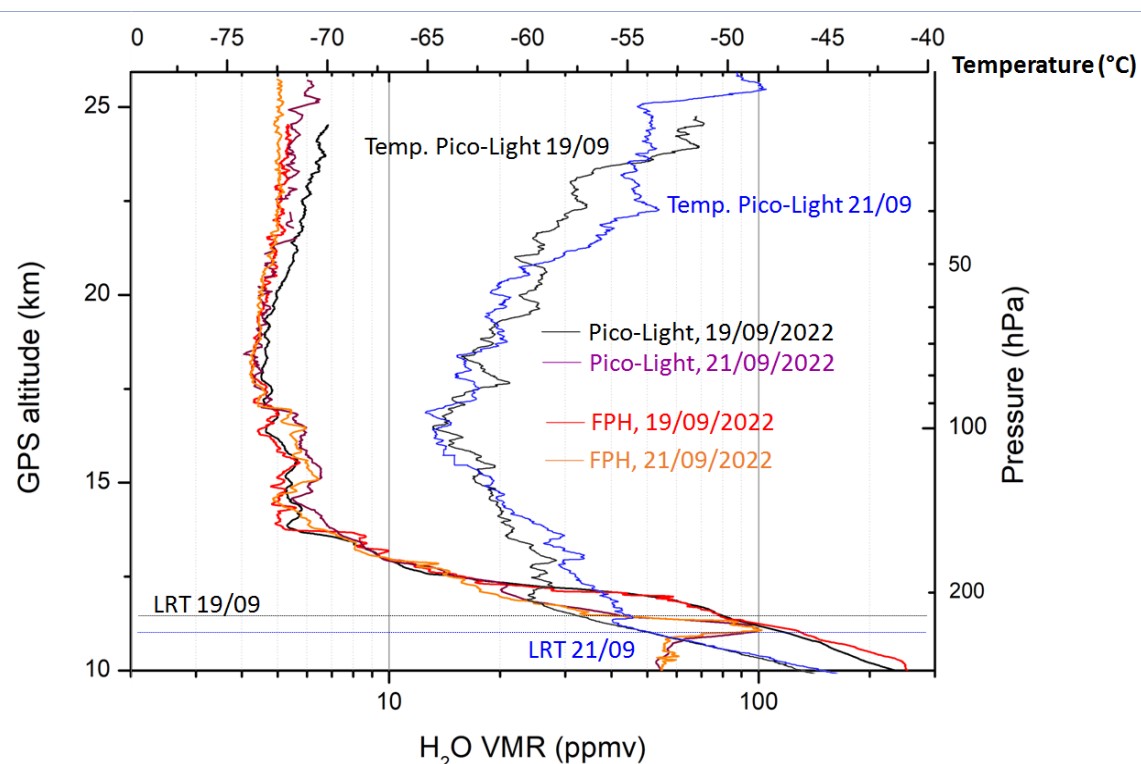

**Figure 6. Vertical profiles of water vapour on September 19 and 21, 2022 from Pico-Light H₂O (descent) and NOAA FPH (ascent). Profiles on September 19 are shown in black (Pico-Light) and red (NOAA FPH). Profiles from the September 21 are shown in grey (Pico-Light) and orange (NOAA FPH).**





On September 21$^{st}$, 2022, the observed hydrated layer between 14 and 17 km is associated with a filamentary structure
originating from the subtropics. The relatively thick (~3km) structure is seen by the ERA 5 reanalysis at pressure levels of
100, 125 and 150 hPa (16.4, 15 and 14 km).

On September 19$^{th}$ and 23$^{rd}$, a large tongue of moist subtropical air was present at 225 hPa (11.4 km) over Aire-sur-l'Adour.
This translates into similar volume mixing ratio at both dates while a difference of about 18 ppmv is observed at 175 hPa
(~13 km). Figure 7 shows vertical profiles from Pico-Light H$_2$O on September 19$^{th}$, 21$^{st}$ and 23$^{rd}$, 2022, compared to maps of
potential vorticity from the ERA5 reanalysis at 175 and 225 hPa, on September 19$^{th}$ and 23$^{rd}$. The balloon position is marked
with a black circle. This large intrusion is seen by ERA 5. At 175 hPa, a thin filament of high potential vorticity is seen by
ERA5 over Aire-sur-l'Adour on the 19$^{th}$ but not anymore on the 23$^{rd}$, which explains the 18 ppmv difference observed
between both dates.

**Figure 7.a) Water vapour vertical profiles from descent measurements of Pico-Light H$_2$O on the 19$^{th}$ (black), 21$^{st}$ (blue) and 23$^{rd}$
(red) of September 2022. Right panels: maps of potential vorticity from the ERA 5 reanalysis at 175 and 225 hPa both on the
19$^{th}$(b and d respectively) and 23$^{rd}$ (c and e respectively). The location of Aire-sur-l'Adour is indicated by a black circle.**

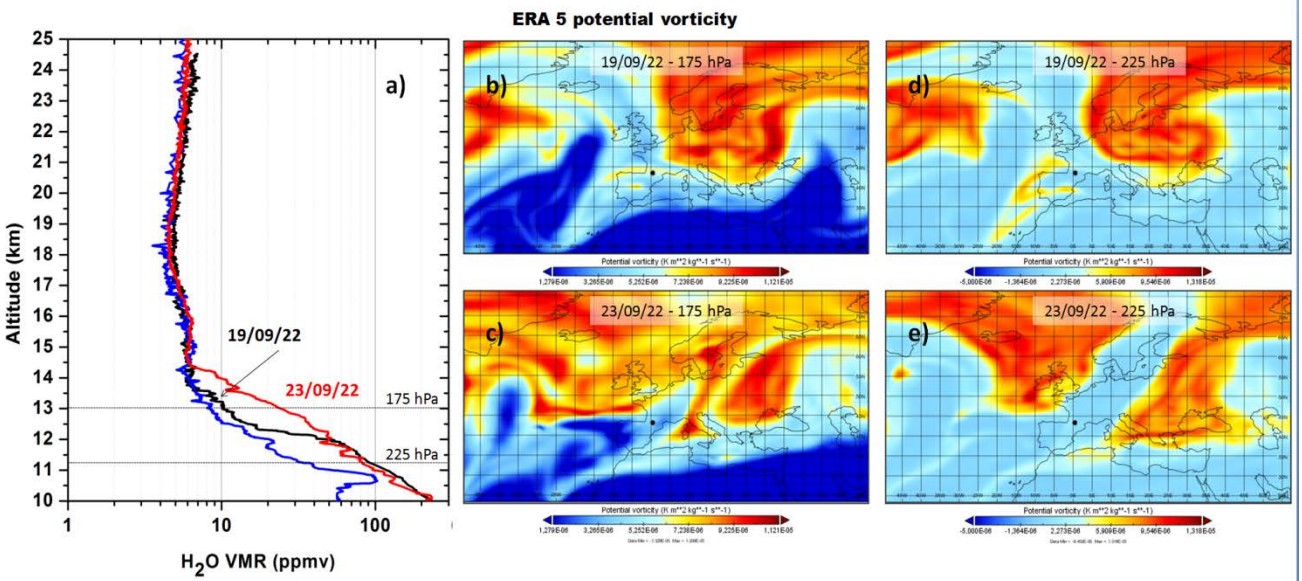

Going further into quantitative comparison, the comparison between Pico-Light H$_2$O and the NOAA FPH has been
performed within three distinct layers between the ground and few kilometres below the balloon burst altitude, where Pico-
Light measurements are free from contamination: the troposphere, spanning altitudes between the ground and the lapse rate
tropopause;  the tropopause region (TR) between the LRT and the CPT, and the stratosphere, defined here by water vapour
mixing ratio between 3.5 and 6.5 ppmv. In this last, we expect the lowest influence from the dynamics in the TR, limiting the
variability of the local water vapour content.





Table 4 summarizes the relative differences between measurements from Pico-Light H₂O and NOAA FPH during three flights of the AsA2022 campaign, when both sensors were launched within a 30 minute time interval. The data from Pico-Light and FPH data are both averaged over bins of altitude of ± 100 m.

The relative difference is calculated such as $\text{Rel. diff} = \left(\frac{\chi_{pic\,o} - \chi_{FPH}}{\chi_{FPH}}\right) \times 100$.

The average standard deviation is calculated such as : $\sigma = \sqrt{\frac{((n_1-1)s_1^2 + (n_2-1)s_2^2 + \cdots + (n_k-1)s_k^2)}{(n_1 + n_2 + \cdots + n_k - \bar{k})}}$

Where $n_k$ is the sample size for the $k^{th}$ group, $s_k$ the standard deviation for the $k^{th}$ group and k the total number of groups.

**Table 4. Mean relative difference and average standard deviations between in-situ water vapour mixing ratio from the Pico-Light H₂O and the NOAA on September 19, 21 and 23, 2022. Volume mixing ratios have been averaged over altitude bins of ± 100 m around Pico-Light and FPH altitudes. Flights have occurred from Aire-sur-l'Adour (France).**


| AsA 2022 | FPH | Pico | Date | Alt range (km) | 19/09/2022 (%) | 21/09/2022 (%) | 23/09/2022 (%) | Mean difference (%) | |
|---|---|---|---|---|---|---|---|---|---|
| **Troposphere** | Asc | Asc | 19/09 | 0.23-11.51 | 4.66 ± 24.06 | 1.43 ± 16.28 | 5.43 ±15.61 | 3.84 | ±23.64 |
| | | | 21/09 | 1.14 - 11.07 | | | | | |
| | | | 23/09 | 1.53 - 12.29 | | | | | |
| **Tropopause** | Asc | Asc | 19/09 | 11.51 - 13.00 | -2.61 ± 4.43 | 5.91 ± 7.55 | -1.32 ± 3.10 | 0.66 | ± 7.30 |
| | | | 21/09 | 11.07 - 12.86 | | | | | |
| | | | 23/09 | 12.29 - 13.13 | | | | | |
| **Tropopause** | Asc | Desc | 19/09 | 11.51 - 16.48 | 1.77 ± 7.94 | 6.73 ± 7.17 | -5.49 ± 5.70 | 1.00 | ± 9.19 |
| | | | 21/09 | 11.07 - 16.35 | | | | | |
| | | | 23/09 | 12.29 - 13.13 | | | | | |
| **Tropopause** | Desc | Desc | 19/09 | 11.51 - 16.48 | 3.57 ± 10.4 | 6.63 ± 8.95 | | 5.10 | ± 11.0 |
| | | | 21/09 | 11.07 - 16.35 | | | | | |
| | | | 23/09 | (no FPH data) | | | | | |
| **Stratosphere** | Asc | Desc | 19/09 | 16.48 - 18 | 3.18 ± 2.01 | 1.92 ± 2.90 | -0.1 ± 5.84 | 1.66 | ± 5.03 |
| | | | 21/09 | 16.35 - 21.38 | | | | | |
| | | | 23/09 | 13.13 - 17.50 | | | | | |
| **Stratosphere** | Desc | Desc | 19/09 | 16.48 - 18 | 4.47 ± 2.73 | | | | |
| | | | 21/09 | (no FPH data) | | | | | |
| | | | 23/09 | (no FPH data) | | | | | |





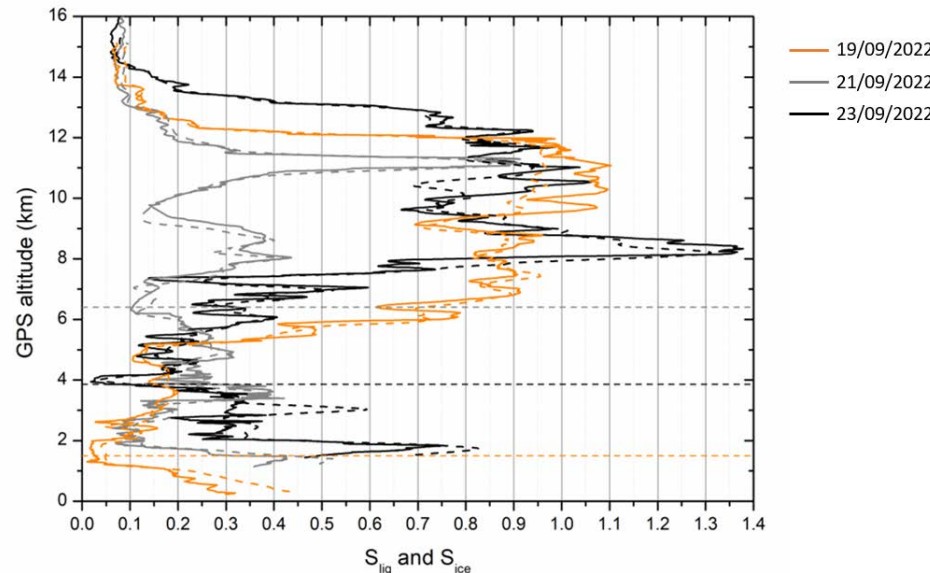

**Figure 8: Flights of September 19 (orange), 21 (grey) and 23 (black), 2022, from Aire-sur-l'Adour (France). Dash horizontal lines show the altitude where ice phase is found on FPH mirror for each flight. Saturation ratios during the ascent are given over ice above this altitude and over liquid water below.FPH saturation ratio are shown as full lines and those from Pico-Light as dashed lines.**

We consider pairs of profiles including ascent and descent profiles for a given day of flight. Since ascent data from Pico-Light suffer from outgassing above 13 to 14km, only tropospheric comparisons are obtained in this case. During the descent, due to the short flight chain, the altitude above which outgassing is affecting measurement is variable from one given flight to another, scaling from 18 to 21.4 km. The saturation ratio from FPH and Pico-Light measurements are shown in Fig. 8. On the flights of the 19[th] and 23[rd], thick saturated layers ($S_{ice} \geq 1$) are found. Particularly, on the 23[rd], the ice saturation ratio

($S_{ice}$) exceeds 1.3 between 8 and 9 km and is associated (not shown here) with a liquid water saturation ratio ($S_{wat}$) near 1.0. Except in this altitude range, the balloon flew through a large layer expanding from 7.5 to 13 km, where the $S_{ice}$ is found close to 0.9, whereas the $S_{liq}$ did not exceed 0.7. In this case, both Pico-Light and FPH have flown through a mixed phase cloud. For this flight, the outgassing contamination is the largest and affect measurements down to 18 km during the descent. This is due to liquid droplets and or ice particles sticking to the balloon and parachute while ascending through the cloud,

which then evaporated or sublimated during the rest of the ascent. Similar behaviour is observed for the flight of the 19[th], but to a lesser extent. In the case of the flight of the 19[th], the balloon flew through a thick layer between 9.5 and 11.6 km, where the ice saturation ratio was near 1.0 and liquid water saturation ratio of 0.6. In this case, both Pico-Light and FPH have flown through a thick ice cloud, as confirmed by EUMETSAT (not shown) cloud top observations (so is the case for September 19[th]). EUMETSAT has observed cloud tops nearby 11.5 km. The situation on the 21[st] is dramatically different, relative

humidity over ice (above 6.4 km) and water (below 6.4 km) is lower than 0.45 except between 11 and 11.3 km, where $S_{ice}$ is



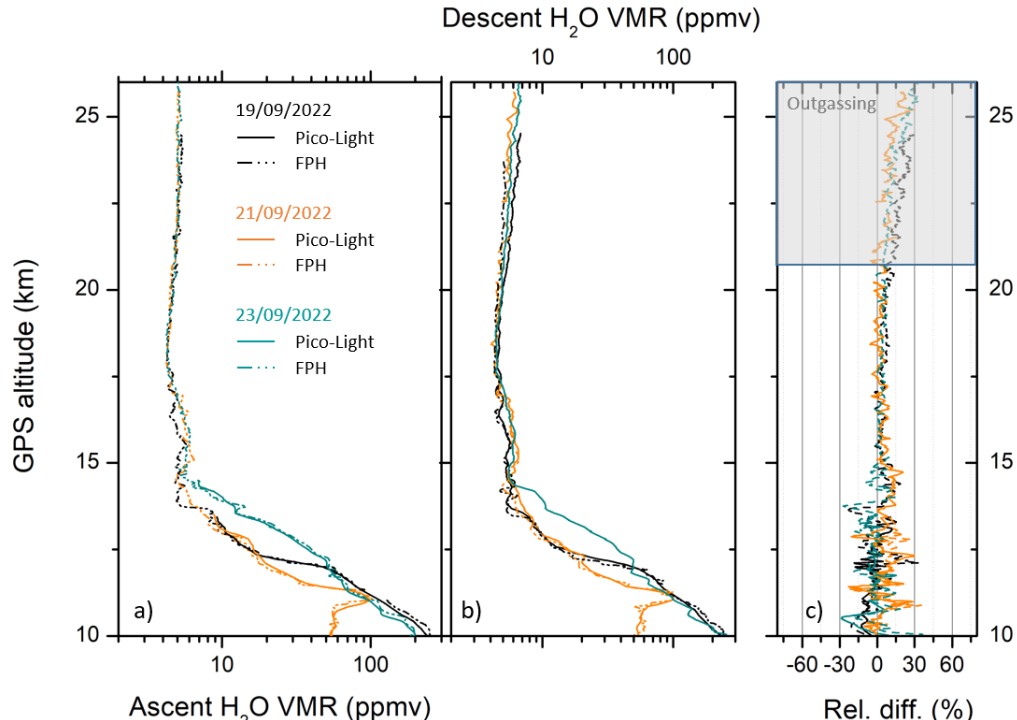

**Figure 9: Panels a and b: vertical profiles of water vapour mixing ratio during the ascent (panel a) and descent (panel b) from Pico-Light (full lines) and FPH (dash lines) on September 19[th], 21[st] and 23[rd], 2022, from Aire-sur-l'Adour (France). Panel c: Relative differences between Pico-Light and FPH during ascent and descent (solid and dash lines respectively).**

of about 0.9. Then, on the 21[st], flights have been realized under clear sky conditions. In this case, the contamination effect of Pico-Light is reduce by a large amount, with a top altitude free of contamination around 21.4 km. In general the NOAA FPH instrument is not affected by outgassing during ascent, except on the 19[th] (from 24.5 km only), since the distance between the parachute, balloon and the instrument is 36 m, 21 m longer than for Pico-Light. For next flights of Pico-Light we intend

to increase the distance between the instrument and balloon/parachute to reduce the potential for contamination.

Considering only portions of the profile which are free of contamination, the differences between both instruments in general follow the same behaviour with rising altitude and are consistent from one flight to another. The differences decrease with altitude, as does the variability of water vapour. Considering pairs of profiles from the same flight segment (i.e. ascent or descent) should reduce the differences.

In the troposphere, we only consider ascent datasets since both instruments were flown under their own balloons. The relative difference in mixing ratio is highly variable, strongly correlated with vertical structures and therefore to the variability of tropospheric water vapour. The mean tropospheric relative differences average is of $(3.84 \pm 23.64)\%$.

The comparison between Pico-Light $H_2O$ and the NOAA FPH above the LRT is illustrated in Fig. 9.





In the TR, for an altitude range between the lapse rate tropopause (around 11.5 km) and cold point tropopause
(around 16.5 km), the mean relative difference is then of about $(1.90 \pm 8.70)$ %, considering all pairs of comparison. In the
stratosphere, the relative difference between Pico-Light $H_2O$ and NOAA FPH is relatively constant. On average, the relative
difference is of about $(2.37 \pm 4.60)$ %, within the total uncertainty of Pico-Light in this altitude range, scaling from 3.5 to 7.5
%. Considering the altitude range from the LRT up to 20 km, the mean relative difference is of $(4.20 \pm 2.70)$%, still within
the Pico-Light total uncertainty. In the TR, largest differences are found slightly above the hygropause, where the random
uncertainty, induced by the spectrum baseline variability, is the largest. In this altitude range, the random uncertainty is of
about 3.8%, thereby being the largest contributor to the observed differences.

The differences found here are in line with other recent published studies. In the UTLS, Singer et al., (2022) compared
aircraft in situ measurements from FLASH (Lyman-α, Sitnikov et al., 2007), ChiWIS (OA-ICOS, Sarkozy et al., 2020) and
FISH (Lyman-α, Meyer et al., 2015) between 14.7 and 20 km during the StratoClim campaign, with mixing ratios varying
from 4 to 10 ppmv. Per flight, the relative differences between the FLASH and ChiWIS hygrometers below 10 ppmv, varied
from -4.9 % to 3.1%. Between FISH and FLASH, in clear sky conditions, relative differences scaled from -10.1 % up to
11.5%. Considering all 6 flights, Singer et al, (2022), reported average relative difference varying from -0.4 to 1.9%.

Kaufmann et al., (2018) reported intercomparisons in the UTLS between in situ aircraft hygrometers during the
ML-CIRRUS campaign: FISH, HAI (Buchholz et al., 2017), SHARC and AIMS(Kaufmann et al., 2016; Thornberry et al.,
2013). The relative difference between each instruments was calculated against a reference value which was calculated using
the average of measurements from a combination of several of these instruments, varying upon atmospheric conditions. In
the range 4 – 10 ppmv, the relative differences varied from $\pm$ 1% (around 10 ppmv) up to $\pm$ 7% (around 5 ppmv) for the
AIMS and FISH instruments, the only ones capable of measuring such low mixing ratios. The global agreement was found
within $\pm$ 15%.

The differences in measured pressure from Pico-Light and iMet-4 radiosonde flown with the FPH stay within $\pm$ 1.2
hPa below the LRT and within $\pm$ 1 hPa above, and do not have significant influence in the water vapour mixing ratio
differences. As discussed in section 3.4 "Uncertainties", the pressure error only as a minor influence in the overall error
budget. Air temperature differences are the largest in the lower troposphere, reaching as high as 4°C in the first 500 m of
altitude, rapidly decreasing down to within the $\pm$ 0.5°C range where it remains constant from 3.5 km. Comparing pairs of
ascent profiles, the temperature differences do not have a visible impact on the observed relative differences. At least, no
clear correlation is found. However, a moderate correlation is found below the LRT while considering pairs of FPH
ascent/Pico-Light descent profiles. This mainly tells that some of the mixing ratio relative differences observed in this case
are related, to some extent, to local variability of air moisture, though it does not preclude instrumental differences. Indeed,
for some fine vertical structures, it remains difficult to estimate the contribution of local moisture variability and
instrumental errors in the mixing ratio differences observed.

Figure 10 shows the correlations between Pico-Light $H_2O$ and the NOAA FPH measurements between 3.5 and
13000 ppmv. The inset focuses on the range 3.5 to 7 ppmv, found above the LRT. Considering the data from the 3 flights,





the linear correlation slope is 1.018 ± 0.002 between 3.5 and 13000 ppmv with an $r^2$ of 0.95. Restricting the comparison to the TR and stratosphere (mixing ratio from 3.5 to 100 ppmv) the linear correlation slope is 1.008 ± 0.002 with an $r^2$ of 0.974.

In Singer et al., (2022), the reported linear correlation slope coefficients were of about 0.930 in the range 2 to 10 ppmv. Restraining our comparison to the same altitude and volume mixing ratio ranges, the Pearson's r coefficient is of 0.975 and the associated $r^2$ coefficient is of 0.998, similar to those reported in Singer et al., (2022) and Kaufmann et al., (2016). In Kaufmann et al., (2018), the reported $r^2$ values y from 0.948 to 0.996, depending on the instrument pair considered and environmental conditions, in the 1 to 1000 ppmv range.

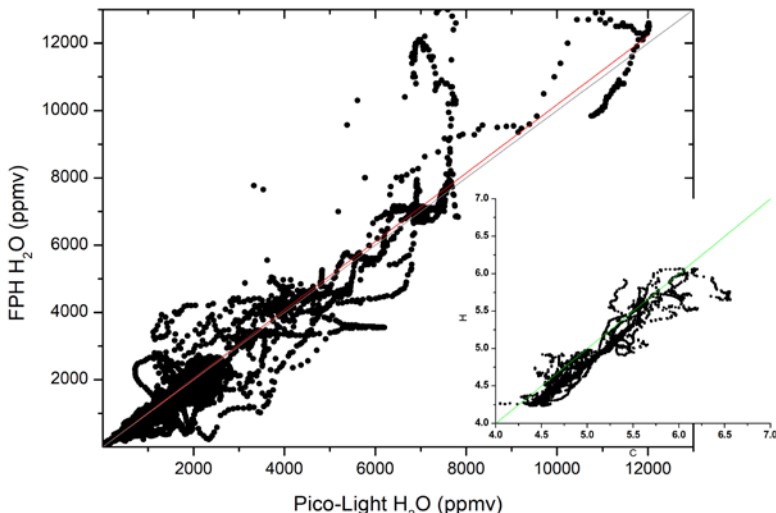


**Figure 10: Correlations between Pico-Light H$_2$O and NOAA FPH retrievals in the range (3.5-13000 ppmv). The inset shows correlations in the range 4-7 ppmv for altitudes higher than 14.5 km.**

**9. Comparisons in the troposphere - relative humidity**

Measurements from Pico-Light H$_2$O enable the calculation of volume mixing ratios (mole fractions) of water

vapour. To compare with meteorological sondes, we calculated relative humidity from Pico-Light data using the Hyland and Wexler (1983) equation for saturation over liquid water and Goff–Gratch (1984) equation for saturation over ice. Both Pico-Light and FPH instruments were flown with an iMet-4 sondes onboard. For each flight of both instruments, a new iMet-4 sonde was used because batteries were drained and sensors were damaged.

A summary of the comparison is given in Table 6 and illustrated in Fig. 11. The RH difference between the sonde and the

Pico-Light or FPH (hereafter, scientific instruments) is obtained such as:

$$\Delta RH = (RH_{sonde} - RH_{inst})$$

In the range 0-7.5 km, RH values measured by iMet-4 sondes compare really well with RH values calculated from FPH and Pico-Light measurements if the T (predominantly) and P measurements used for the calculation of RH come from





the IMet sonde onboard. Larger discrepancies are found otherwise. Then, in the frame of the RH comparisons, the RH values calculated for Pico-Light are obtained using the IMet-4 P and T measurements onboard. In the altitude range from 0 to 7.5

km, the mean difference between iMet-4 and FPH is of -1.2% RH and is of -3.2% RH between IMet-4 and Pico-Light. Comparing Pico-Light and FPH, the average difference is 0.5% RH in the same altitude range. Expanding the altitude limit to 13 km, the average difference becomes $-0.2 \pm 0.7$% RH.

Above 7.5 km, the discrepancies in RH between IMet-4 and the scientific hygrometers increase up to 50% RH nearby 12 km. Above, IMet-4 measurements begin to show unrealistic values. Differences are noticeable between the

comparisons with FPH and Pico-Light, mainly attributed to the implementation of RH corrections in iMet sondes onboard FPH.

M20 sondes from MeteoModem were launched at Aire-sur-l'Adour a few minutes apart from other instruments. Figure 12 shows a comparison of RH measurements from M20 to Pico-Light and FPH. M20 sondes perform well up to about 13 km, and stay within the $\pm$ 10% RH range of differences except for some exceptions where saturation values are found

(e.g. at 8 km on the 21[st]) where M20 sondes underestimate the RH by about 20%. On average, the differences between M20 and FPH between 1 and 13 km is of about $(3.1 \pm 1.1)$ % RH. The mean difference with Pico-Light is of $(3.3 \pm 1.8)$ % RH, within the stated uncertainty. Then, M20 have a small wet bias of about 3% RH on average, compared to the scientific instruments and a dry bias of about 20% RH in case of saturation.

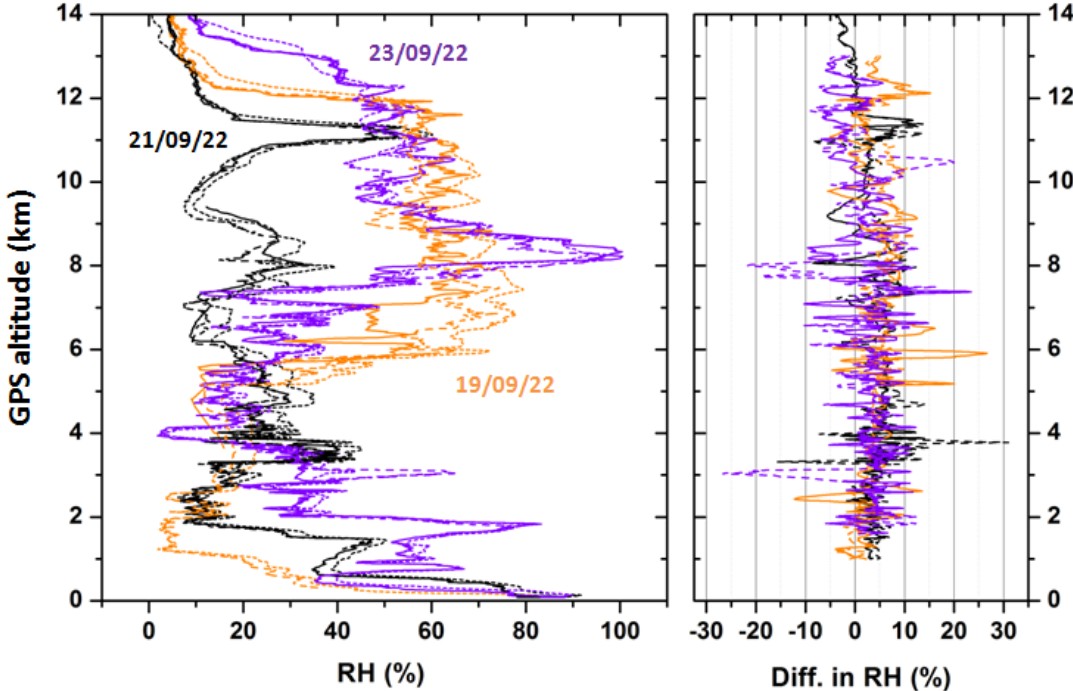

**Figure 11: On the left: Comparison between relative humidity profiles from FPH (solid line: black: Sept. 21[th], orange: Sept. 19[st], purple: Sept. 23[rd]), Pico-Light (dash lines, same color as FPH), and M20 (double dash lines) in the altitude range from 1 to 13 km. On the right: Differences in RH between M20 and FPH (solid line) and between M20 and Pico-Light (dash line).**



In 2011, in the frame of the DEMEVAP experiment (Bock et al., 2013), M10 sondes (predecessor of M20) were
compared to Vaisala RS92 sondes, with a reported mean difference, constant in the same altitude range, of about -6.41%.
More recently, in the frame of the MétéoSwiss intercomparison campaign, held in August-September 2022, the M10 sondes
were compared to Vaisala RS41 with differences less than 3% RH, similarly to the present study (World Meteorological
organization and Intergovernmental oceanographic commission, 2021).

**Table 6: Relative differences in relative humidity between meteorological sondes iMet-4 and M20 and standard deviations,
compared to Pico-Light and FPH retrievals from ground to 13 km.**

| Date of flight | | FPH/ iMet-4 (%) | FPH/M20 (%) | Pico-Light/iMet-4 (%) | Pico-Light/M20 (%) | Pico-Light/FPH (%) |
|---|---|---|---|---|---|---|
| **19/09/2022** | 0-7.5 km | -2.0 ± 4.0 | 4.4 ± 5.4 | -3.3 ± 4.8 | 3.0 ± 2.4 | 1.3 ± 5.1 |
| | 0-13 km | | 4.1 ± 4.8 | | 4.4 ± 3.0 | -0.3 ± 4.8 |
| **21/09/2022** | 0-7.5 km | -0.4 ± 1.6 | 4.5 ± 2.6 | -2.1 ± 2.1 | 5.0 ± 3.9 | -0.4 ± 4.2 |
| | 0-13 km | | 3.4 ± 3.3 | | 4.3 ± 4.0 | -0.8 ± 3.6 |
| **23/09/2022** | 0-7.5 km | -1.2 ± 3.3 | 3.8 ± 4.2 | -4.2 ± 4.4 | 2.4 ± 5.5 | 0.5 ± 7.1 |
| | 0-13 km | | 1.9 ± 4.8 | | 1.3 ± 6.3 | 0.6 ± 7.5 |

## 10. Conclusions

Pico-Light H$_2$O is an in situ tunable diode laser hygrometer developed during 2017 – 2019 to measure water vapour intended
primarily in the upper troposphere and the stratosphere though demonstrating good performances in the troposphere. It is the
lightweight successor of the former Pico-SDLA H$_2$O hygrometer. Pico-Light H$_2$O was flown from the Aire-sur-l'Adour
CNES balloon facility six times between 2019 and 2022. In the frame of the AsA2022 campaign in 2022, the hygrometer has
been compared to in situ measurements by the NOAA FPH, resulting in a mean difference of (2.37 ± 4.60) %, in water
vapour volume mixing ratio in the stratosphere (mixing ratio below 6.5 ppmv), within the retrieval uncertainty of Pico-Light
in this altitude range. In the tropopause region (mixing ratio between 7 and 100 ppmv), the mean relative difference is then
of about (1.90 ± 8.70) %. Tropospheric comparisons reveal a (3.84 ± 23.64)%. mean difference (mixing ratio above 100
ppmv) and a mean difference in calculated RH values of about (-0.2 ± 0.7) % RH. During this campaign, iMet-4 radiosondes
were installed on all balloons carrying the Pico-Light and FPH instruments. M20 sondes were launched on their own
balloons within 30 minutes of the in situ instruments and a comparison of tropospheric RH values from Pico-Light H$_2$O and
FPH has been performed. Between 0 and 7.5 km altitude, iMet-4 sondes agreed to within ± 2% RH with FPH and to within ±
4.2 % RH with Pico-Light. M20 sondes are wet biased to about 3% RH compared to FPH and Pico-Light and underestimate
RH by about 20% RH in case of saturation. However, the agreement up to 13 km remains impressive.



## 10. Acknowledgement

This work is based on observations with Pico-Light H$_2$O under a balloon operated by CNES, under the agreement between CNES and CNRS/INSU, within the WP11 of the European project HEMERA H2020 (Integrated access to balloon-borne platforms for innovative research and technology). The CNES staff at Aire-sur-l'Adour is highly acknowledged for its efficiency and valuable support in the organization and performance of the AsA2022 campaign. The Aerodrome Staff is also acknowledged for granting access to their facility, thereby permitting the launch of the NOAA FPH balloons very close to

the CNES balloon launch facility in Aire-sur-l'Adour. Time and effort of the FPH team that participated in this campaign was supported by the NASA Upper Atmospheric Composition Observations program. Finally, we thank the DIRSU Drones from CNRS for their support in the process to obtain flights authorizations from Aire-sur-l'Adour.

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
