# Peer review of "Pico-Light $H_2O$ : Intercomparison of in situ water vapour measurements during the AsA 2022 campaign."

_Atmospheric Measurement Techniques, 2023_

## Referee Comment (RC1)

**Review of: "Pico-Light H2O: Intercomparison of in situ water vapour measurements during the AsA 2022 campaign." by Ghysels et al.**

**General comments**

The manuscript is well written and structured and presents an impressive instrument development and intercomparison. The topic and the quality of the manuscript fit very well into the scope of AMT. However, I still have some comments or questions about the manuscript that should be answered before publication. However, only minor changes to the manuscript are necessary. In general, it would also be good for the manuscript to be read by a native speaker. There are still some minor linguistic errors in the text which should be corrected before publication.

**Specific comments/questions:**

- Lines 70-73: Here, the rather good agreement of the PICO-SDLA is mentioned in comparison to the other intercomparison campaigns. It should me mentioned in the manuscript that in your study from 2016 only **mean differences averaged over large height ranges** of several kilometers where used. The other studies cited here used mostly a direct comparison on a high resolution (mostly on 1Hz), which automatically leads to a larger discrepancies from time to time. If you would do it in the same way (high resolution) for the PICO-SDLA you would find similar deviation around 10% as reported by others. Therefore this comparison is not fair.

- Lines 125-126: " Like all optical absorption hygrometers": That's a bit too general. For example, there are absorption hygrometers that measure at a wavelength of 1.4mum and are not sensitive to small concentrations of water vapor. I recommend replacing the "all" with "many".

- Line 167-168: It should be better explained that PICO-Light is using two absorptions lines for different atmospheric conditions and also why (Just with one sentence). Otherwise, it is not clear why the electronic should switch to another absorption line. You wrote the detailed explanation in Section 3.4

- Line 175: It is not clear to me, why you do not see any outgassing of the electronics during ascent of the balloon. The laser is installed downstream and there could be potentially some mixing of artificial water vapor into the atmospheric air volume. I agree for the descent of the balloon, that the measurements are not

affected by outgassing. If you just measuring during descent, you should probably better mention that in the text. In addition, you mentioned in lines 368-369 that PICO-Light provides reliable measurements during ascent and decent. This brings me to the question, if the uncertainties in Table 2 are the same for ascent and descent ?

- Line 333-342: This paragraph could be potentially removed because it is not essential for this particular study as no MLS data are used here.

- Sections 4 and 5: This two sections describe the two other instruments used for the intercomparison to the PICO-Light. It would be helpful for the reader to also state the final water vapor uncertainties of both instruments in the text. This allows to better interpret the results from the comparison and would complete the instrument description in terms of performance.

- Line 427: Meteorological data are here mentioned the first time without any explanation. There should be at least a very brief introduction of ERA5 in text before.

- Line 435-436: The relatively thick water structure is mentioned to be seen in the ERA5 data. But there are no ERA5 data shown. So I would recommend to state (not shown) somewhere in this sentence.

- Line 471-472: I do not understand why the water vapor profile is affected by outgassing during descent. As you mentioned before outgassing should not be a problem for the measurements during descent.

- Figure 8: Which saturation mixing did you choose in case of the PICO-Light ? Is the entire profile showing $S\_ice$ or $S\_liq$ or was the same procedure applied as for the FPH ?

- Line 476-479: Actually I cannot really believe that in the entire altitude range of 7.5 to 13 km you will still find a mixed-phase cloud with both $S\_ice$ and $S\_liq$ below 1. In case there are ice particles actually all droplets would quickly evaporate and increase the size of the ice particles. In addition below -38°C all liquid droplets would freeze instantaneously and form ice particles. And typically ice should not yield to large contamination. I guess the contamination is just coming from the small layer where you reach also the $S\_liq$ close to 1. There I agree that this might be a mixed phase cloud. Could be from convective origin ? Another thing which I do not understand is, why you still see contamination of during descent down to 18km? Is it because of the optical cell is contaminated by water ?

- Lines 545-547: Can you please explain how you calculated RH in part of the profile for each instrument/sonde. The radiosonde typically just give RH wrt. water for the entire profile. When comparing to the PICO-Light or FPH the mixing ratio should also be converted just to RH wrt. water. So you actually do not need to calculate RH wrt. ice using the Goff–Gratch equation for the comparison.

**Technical comments/suggestions:**

- The space between numbers and units appears sometimes to large and both are in additional sometimes wrapped in the text at the end of the lines.

- Line 41: Please write units without a dot in between: +0.3 W.m-2 .K-1.

- Line 64: (Rollins et al., 2014) compares aircraft- and balloon-based... You should cite studies incorporated in the text without brackets. There are several places in the text were the citation style should be changed like line 67 or line 70. The bracket citation style should only be used, if studies are not incorporated in the sentence like you did in line 35 for example.

- Line 72: I suggest to rephrase "multiplying" by "increasing the amount of "

- Line 78-79: "where absolute modulation of the local mixing ratio scales are within 10 to 20% of the typical mixing ratio." I suggest to skip the "are".

- Line 88: "meteorological sonde" should be plural.

- Line 127: "at the difference of other" should be replaced by "in contrast to other".

- Line 291: I suggest the word "of" here: "uncertainty is of reduced to 0.1%".

- Line 296: I suggest to change to: "From about 15 km downward,"

- Table 2: I suggest to put the "hygro" flag on one of the first two columns. Or is there a reason for choosing the random error to flag ?

- Line 366: Skip the word "sondes".

- Figure 4: Is the seems that FPH flight train also contained an ozone sonde ? Or what is the white box between radiosonde and FPH ? I suggest to either label it or remove it from the schematics.

- Line 408: ", the only the mechanical" Please skip "the" after the comma.

- Figure 6: PICO-Light profile from the September 21 is not shown in grey. It is shown in purple. Please change the caption here.

- Figure 7: I recommend to improve the scaling of the PV maps. All color-codes should range from the same values. I suggest to go from 1 to 13 PVU for all four plots.

- Line 454: "In this last". Please add here "range" before the comma.

- Line 487: "reduce" -> "reduced"

- Line 538: "y" ? Maybe it should mean "range"

- Figure 10: It would be great, if you could include the regression and r2 values in each panel.

- Line 547: "sondes" -> "sonde"

- Figure 12 is missing !

---

## Author Response (AR1)

Reponse to reviewer #1: manuscript 2023-191
Pico-Light H$_2$O: Intercomparison of in situ water vapour measurements during the AsA 2022 campaign.

**General comments**
The manuscript is well written and structured and presents an impressive instrument development and intercomparison. The topic and the quality of the manuscript fit very well into the scope of AMT. However, I still have some comments or questions about the manuscript that should be answered before publication. However, only minor changes to the manuscript are necessary. In general, it would also be good for the manuscript to be read by a native speaker. There are still some minor linguistic errors in the text which should be corrected before publication.

**Specific comments/questions:**
**RV#1 :** Lines 70-73: Here, the rather good agreement of the PICO-SDLA is mentioned in comparison to the other intercomparison campaigns. It should me mentioned in the manuscript that in your study from 2016 only mean differences averaged over large height ranges of several kilometers where used. The other studies cited here used mostly a direct comparison on a high resolution (mostly on 1Hz), which automatically leads to a larger discrepancies from time to time. If you would do it in the same way (high resolution) for the PICO-SDLA you would find similar deviation around 10% as reported by others. Therefore this comparison is not fair.
**Author:** In fact, in the paper Ghysels et al., AMT, 2016 the comparison has been calculated in the same way as the referred papers: a direct comparison on a high resolution vertical scale. The values given for comparison over layers of several kilometers are the averaged difference over the range 15-23 km.  I will add the standard deviation (I forgot to add) to the mean difference so anyone can see the deviation. Then, the sentence becomes: " The differences between both instruments was scaling from (0.5± 4.5)% to (1.9±9.0) % (25 to 100 ppbv) above the cold point tropopause."

Lines 125-126: " Like all optical absorption hygrometers": That's a bit too general. For example, there are absorption hygrometers that measure at a wavelength of 1.4mum and are not sensitive to small concentrations of water vapor. I recommend replacing the "all" with "many".
**Author:** Thank you, it is done.

Line 167-168: It should be better explained that PICO-Light is using two absorptions lines for different atmospheric conditions and also why (Just with one sentence). Otherwise, it is not clear why the electronic should switch to another absorption line. You wrote the detailed explanation in Section 3.4
**Author:** Noted. We added this sentence to clarify: "Pico-Light uses two absorption lines for the sounding: one is used in the lower troposphere (the weakest one), and the second (having a line intensity ten times larger) from higher in the atmosphere in order to compensate for the water vapour mole fraction change."

Line 175: It is not clear to me, why you do not see any outgassing of the electronics during ascent of the balloon. The laser is installed downstream and there could be potentially some mixing of artificial water vapor into the atmospheric air volume. I agree for the descent of the balloon, that the measurements are not affected by outgassing. If you just measuring during descent, you should probably better mention that in the text. In addition, you mentioned in lines 368-369 that PICO-Light provides reliable measurements during ascent and decent. This brings me to the question, if the uncertainties in Table 2 are the same for ascent and descent ?
**Author:** In fact , we do have contamination during the ascent (which starts to be detectable from 14 km of altitude) most probably a combination of outgassing from the electronics and balloon, flight chain

elements. We can discriminate the origin of the contamination comparing ascent and descent polluted measurements: during ascent the contamination is much larger (balloon). I slightly changed the sentence, such as: " The optical cell, shown in Fig. 1, includes the 1-m open structure beneath the electronic enclosure." as well, the sentence line 370-374 has been changed such as " The flight chain separates the instrument from the balloon by 17 m. In this case, the ascent measurements become affected by outgassing from the balloon envelope and other elements on the flight chain (major source) and instrument electronics (minor) at about 14 km. Below, measurements from the ascent and the descent of Pico-Light agree to within 3 %."
Lines 369-370: The sentence " The Pico-Light hygrometer is able to provide reliable measurements during the balloon ascent (up to about 14 km) and the descent under parachute." has been corrected.

Line 333-342: This paragraph could be potentially removed because it is not essential for this particular study as no MLS data are used here.
**Author:** This paragraph was intended to pose the context of the NOAA instrument, as a reference here, showing reliability for a long time now. I would leave it as is.

Sections 4 and 5: This two sections describe the two other instruments used for the intercomparison to the PICO-Light. It would be helpful for the reader to also state the final water vapor uncertainties of both instruments in the text. This allows to better interpret the results from the comparison and would complete the instrument description in terms of performance.
**Author:** The uncertainties for the NOAA FPH instrument have been given lines 268-270. " By comparison, frost control instabilities of the NOAA FPH dominate the measurement uncertainty budget ($\pm 2\sigma$), ranging from $\pm 10\%$ in the lower troposphere to $\pm 2\%$ ($\pm 100$ ppbv) in the stratosphere (Hall et al., 2016)."
About M20: the technical specifications state 3% RH as an absolute accuracy. It has been added to the text, line 351.

Line 427: Meteorological data are here mentioned the first time without any explanation. There should be at least a very brief introduction of ERA5 in text before.
          **Author:** Thank you for pointing it. Therefore, some precisions have been brought lines 426-430: "An exception is found at 15.6 km where a thin 500 m-thick hydrated layer is also visible on the 19[th], again by both instruments at the same altitude, which is not seen in European reanalyses, ERA 5 (Fig. 7) due to ERA5 coarse vertical resolution. ERA5 is the latest climate reanalysis produced by ECMWF, providing hourly data on 137 vertical levels. here, we use potential vorticity as a dynamical tracer in the upper troposphere and stratosphere."

Line 435-436: The relatively thick water structure is mentioned to be seen in the ERA5 data. But there are no ERA5 data shown. So I would recommend to state (not shown) somewhere in this sentence.
**Author:** Done, thank you.

Line 471-472: I do not understand why the water vapor profile is affected by outgassing during descent. As you mentioned before outgassing should not be a problem for the measurements during descent.

**Author** : My guess would be that, first, we cross the old balloon plume at the beginning of the descent and/or secondly, the cell structure could be polluted due to the closeness with the balloon during the ascent. Then, it may take a little time to get rid of the contamination, which become not detectable below 18 to 20 km of altitude.

We are planning additional flights in 2024, comparing with other instruments. We hope to have the agreement to launch longer flight chains: if so, we will be able to test whether we still see such contamination during the beginning of the descent or not.

Figure 8: Which saturation mixing did you choose in case of the PICO-Light ? Is the entire profile showing S_ice or S_liq or was the same procedure applied as for the FPH ?
Author: we revised this section. Figure 8 has been revised, showing Pico RH and RHi, together with the SRH curve to help in the interpretation.

[Figure]

**Figure 8: Profiles of RH (black) and RHi (grey) for the flights of September 19, 21 and 23, 2022, from Aire-sur-l'Adour (France). Dash lines are the calculated saturation relative humidity (SRH).**

Line 476-479: Actually I cannot really believe that in the entire altitude range of 7.5 to 13 km you will still find a mixed-phase cloud with both S_ice and S_liq below 1. In case there are ice particles actually all droplets would quickly evaporate and increase the size of the ice particles. In addition below -38°C all liquid droplets would freeze instantaneously and form ice particles. And typically ice should not yield to large contamination. I guess the contamination is just coming from the small layer where you reach also the S_liq close to 1. There I agree that this might be a mixed phase cloud. Could be from convective origin ? Another thing which I do not understand is, why you still see contamination of during descent down to 18km? Is it because of the optical cell is contaminated by water ?

Author : We revised this paragraph, refining our description and analysis.
"On the flights of the 19th and 23rd, thick saturated layers are found (RH or RHi>SRH). Particularly, on the 23rd, between 8 and 9 km, a layer of RH>SHR is found. Additionally, a thick layer of air saturated over ice expands from 8 to 13 km, though bringing less contamination compared to the case of RH saturation (found between 8 and 9 km). In this case, both Pico-Light and FPH have flown through a mixed phase cloud between 8 and 9 km. For this flight, the outgassing contamination is the largest during the ascent and affect measurements down to 20 km during the descent. This is due to liquid droplets and or ice particles (to a lesser extent) sticking to the balloon and parachute while ascending through the cloud, which then evaporated or sublimated during the rest of the ascent. Similar behaviour is observed for the flight of the 19th, but to a lesser extent. In the case of the flight of the 19th, the balloon flew through a thick layer between 9.5 and 11.6 km, where the RH is above SRH. Between 6 and 12 km, the RHi is above SRH (RHi>100%). In this case, both Pico-Light and FPH have flown through a thick ice cloud within which

mixed phase layers are found. The presence of such thick cloud is confirmed by EUMETSAT (not shown) cloud top observations (so is the case for September 19th). EUMETSAT has observed cloud tops nearby 11.5 km. The situation on the 21st is dramatically different, only a thin layer where RHi is above SRH (altitude between 11 and 11.3 km). Then, on the 21st, flights have been realized under clear sky conditions."

Response to reviewer #2: manuscript 2023-191

Review report

Pico-Light H2O: Intercomparison of in situ water vapour measurements during the AsA 2022 campaign

By Ghysels et al.

This paper presents intercomparison results of balloon borne Pico-Light H2O tunable diode laser hygrometer with NOAA frost point hygrometer and radiosonde relative humidity sensors over France in September 2022. Overall, Pico-Light $H_2O$ shows very good results from the lower troposphere to the lower stratosphere, which are impressive. The manuscript is basically well written, but there are some places where revisions and corrections are needed. I think that the paper will be acceptable for publication in Atmospheric Measurement Techniques after considering my comments and suggestions written below.

Major comments

(1) Figures for saturation ratio (Figure 8) and relative humidity (Figures 11 and 12)

First of all, I am afraid that the current Figure 11 should be actually "Figure 12" for M20, and Figure 11 for iMet-4 is missing.

Author: This is a mistake, thank you for bringing this up. Corrections have been applied. The comparison with Imet-4  is only shown in the table 6.

Another concern is regarding the caption of Figure 8, "Dash horizontal lines show the altitude where ice phase is found on FPH mirror for each flight. Saturation ratios during the ascent are given over ice above this altitude and over liquid water below." Before explaining my concern, let me confirm that saturation ratio (in unit 1) is equivalent to relative humidity (in unit %) (and Figure 8 and Figures 11/12 are somewhat redundant). Am I correct?

Author: Yes, they are somewhat redundant since only the M20 measurements are missing in figure 8 as well as the relative difference panels.

Changing relative humidity (or saturation ratio) definition depending on the phase of condensate on the chilled mirror of NOAA FPH is not very appropriate for the purpose here. The phase of the condensate is nothing to do with the environmental conditions whether water vapor could be condensed as liquid water or ice. Instead, the two phases of the condensate should be viewed as just two different "sensors" for the atmospheric water "vapor" pressure. If the condensate on the mirror is liquid (ice), then we use the Clausius-Clapeyron equation for liquid (ice) to calculate the water vapor pressure values. Thus, we

need to know the phase of condensate when converting NOAA FPH temperature values to water vapor pressure values, but changing the definition of RH/S (i.e. which is to be used for the denominator, liquid or ice version of CC eq.) depending on the FPH condensate phase is nonsense. Probably, the best way is to use always RH (or S) over liquid, as done for "Figure 11" following the convention applied for radiosonde RH sensors, and add ice saturation curve for atmospheric temperatures below 0 deg. C in the figure. In this way, we can easily see whether the upper tropospheric wet layer is close to ice saturation (e.g. when it is close to the ice saturation curve) or even close to water saturation (when it is close to 1 or 100%). Section 4 and Fig. 1 of the paper by Fujiwara et al. (2003) may be useful for these calculations and an example of ice saturation profile. (Note that the comparison results, i.e. relative differences, would not be affected by this revision.)

Fujiwara, M., M. Shiotani, F. Hasebe, H. Vömel, S. J. Oltmans, P. W. Ruppert, T. Horinouchi, and T. Tsuda (2003), Performance of the Meteolabor "Snow White" chilled-mirror hygrometer in the tropical troposphere: Comparisons with the Vaisala RS80 A/H-Humicap sensors, Journal of Atmospheric and Oceanic Technology, 20, Issue 11, 1534-1542.

Author: Thank you for this suggestion. We changed the figure 8 to the present one, in order to make easier the interpretation:

[Figure]

The paragraph has been revised accordingly (see lines 473-488).

(2) The average standard deviation at line 459 and in Tables 4 and 5 and in relevant text

First, please explain what do you mean by "group" at line 460, perhaps by showing some actual examples (e.g. how many group members, and what is each member). Then, please explain how we understand the value, (mean difference) +/-(value). Probably, there is a theoretical explanation from Statistics. (e.g. the mean difference is whether statistically significant at a certain confidence interval? Or standard error of the mean?) And/or, in reality, with which values we can compare these values? (By the

way, I note that the authors take 200 meter vertical averages for both instruments to reduce random noises. So, the "value" above should be something after part of random noises has been removed.)

Author: The term "group" means the data comprised within the ± 100 m interval. Then, the standard deviation which is calculated here correspond to the standard deviation of water vapour mole fractions within this ± 100 m layer. In the text, I have clarified this such as (line 463): " Where $n_k$ is the sample size for the $k^{th}$ group, which includes the data within the ± 100 m interval, $s_k$ is the standard deviation of water vapour mole fractions within the $k^{th}$ group and k the total number of groups."
We proceed with the ± 100m averaging to smooth out small scale variabilities, especially in the troposphere, that may arise from tropospheric dynamics.  The Pico-Light and FPH have not been flown under the same balloon and then, tropospheric relative differences are mostly due to small scale moisture variations. Such averaging (over a short altitude range) in the stratosphere is not expected to have a large impact in the differences reported, while in the tropopshere it helps reducing the influence of external perturbations.

The paragraph has been revised such as:
" Table 4 summarizes the relative differences between measurements from Pico-Light $H_2O$ and NOAA FPH during three flights of the AsA2022 campaign, when both sensors were launched within a 30 minute time interval. The data from Pico-Light and FPH data are both averaged over bins of altitude of ± 100 m. In highly stable stratosphere, such averaging is not expected to significantly influence the comparison. In the case of tropospheric comparison, it is intended to reduce the influence of environment variability on the comparison. Indeed, Pico-Light and FPH have been flown under separated balloons: in such case, the tropospheric dynamics induces small scale mole fractions variabilities which have an impact on the comparison."

" $\chi_{pico}$ and $\chi_{FPH}$ are the average mole fractions from Pico-Light and FPH over the ± 100 m altitude bins. The $n_k$ is the sample size for the $k^{th}$ group, which includes the data within the ± 100 m interval, $s_k$ is the standard deviation of water vapour mole fractions within the $k^{th}$ group and k the total number of groups."

(3) Figures 6, 8, 11, and 12 (although 11 or 12 is missing)
Please prepare different panels for different days. Currently, different days are mixed in one panel, so that the characteristics of the profiles are not easy to grasp.

Author: Noted.
The figures have been updated accordingly such as:

[Figure]

Figure 6: Vertical profiles of water vapour on September19, 21 and 23, 2022 from Pico-Light H2O (black line) and NOAA FPH (red line).

[Figure]

Figure 8 : Profiles of RH (black) and RHi (grey) for the flights of September 19, 21 and 23, 2022, from Aire-sur-l'Adour (France). Dash lines are the calculated saturation relative humidity (SRH).

(4) In Abstract and Introduction

Please add the keywords "tunable diode laser hygrometer" and "open cell" to characterize the Pico-Light H2O instrument. I think these two are key for this instrument, but such information appears only later in Section 3. Probably, also add the information on the two water vapor vibrational-rotational lines and the fact that one is used for tropospheric measurements and the other for stratospheric measurements.

Author: Done. In the abstract, we changed : " **Abstract.** The mid-infrared lightweight tunable diode laser hygrometer hygrometer, "Pico-Light H2O",..."

As well, in the introduction : " In this work, we report the development of a rugged, lightweight, open-cell, tunable diode laser hygrometer,..."

(5) Description of the instrument details in Section 3

Section 3 is basically very well written. But, adding some information may be useful. Please consider the list below:

- Table 1 at laser wavelength: Could you also add the range of the wavelength and the wavelength resolution? (I am referring to the right bottom box in Figure 2.) Typical values would be enough.

Author : We typically have a $8.10^{-4}cm^{-1}$ resolution per spectrum, so about 2.10-4 µm (spacing between two consecutive datapoints on a given spectrum). The diode laser is tunable over several cm-1. In table 1, I specified 2.63±0.05, where ±0.05 is the range between the two absorption line used.

- Is my understanding correct that water vapor data from Pico-Light H2O can be obtained only when the instrument, in particular the micro SD shown in Figure 2, is safely recovered? Is this fact clearly written somewhere in Section 3? If not, please write it.

Author: Right, at the difference of former Pico-SDLA for which we do have TM/TC onboard. In this last case, we can to send orders to the instrument and download data while on flight. For Pico-Light, we only rely on the instrument recovery and the instrument operates fully standalone. We modified the text line 172-173 such as :"Essential diagnostics and information about the physical environment are stored on a micro-SD drive along with the data, which then require to recover the instrument following landing."

- Is this instrument capable of daytime flights? In other words, aren't there solar light contaminations to the spectra?

Author: Yes, the instrument can be operated during daytime. The only change would be that the air temperature reading can be affected in some amount, leading to a minor increased uncertainty while processing the atmospheric spectra (spectroscopy related).

- Lines 189-191: I do not understand what this sentence means. Please add more explanations, e.g. to "the lower of the two values", "corresponding sensor".

 Author: Thank you, it looks like the sentence was not well written. It has been revised such as: " The temperature value used to analyze the spectra was the coldest one since it was assumed to be less affected by solar radiation."

Indeed two temperature sondes are attached to the instrument (line 183). We pick the coldest one as an input for the spectra processing.

Minor comments

- Line 17 (and lines 393-394) – please add the information of longitude and latitude for these sites.

Done.

- Line 57: GCOS is Global Climate . . .

Corrected, thank you.

- Lines 61-73: There are several instrument names here. Probably add the information on the methodology. For example, "Lyman-alpha hygrometer FLASH-B", "NOAA/CMDL frost point hygrometer", etc.
Done thank you. The text has been revised accordingly:

-Lines 62-63: "%). In Vömel et al. (2007), the comparison between Lyman-alpha hygrometer FLASH-B and the NOAA/CMDL frost point hygrometers in the stratosphere lead to..."

-Lines 67-69: "(Kaufmann et al., 2018) reported stratospheric comparisons of Lyman-alpha FISH and AIMS linear quadrupole mass spectrometer to an average reference value calculated from measurements from AIMS, FISH, the tunable diode laser spectrometers SHARC and HAI."

- Line 64: (Rollins et al., 2014) - - > Rollins et al. (2014). Make the similar correction also for lines 70-71.

- Line 184: This is probably Nash et al., 2011, https://library.wmo.int/idurl/4/50499, which is missing in the reference list.

Thank you, right.

- Lines 545-546: Hyland and Wexler (1983) and Goff-Gratch (1984) – these are not in the reference list.

The references have been added, thank you.

- Lines569-574: Information on M10 radiosonde is probably not relevant to this manuscript.

We simply removed this paragraph.

- (Regarding the contamination: It seems to me that the electronics box located just above the optical cell would be a big source of contamination, as the box would collect cloud particles when the instrument goes through cloud layers in the troposphere and as it is much closer to the cell compared to the parachute and balloon. Isn't it possible to rotate the instrument by 90 degree when it is connected to the balloon+parachute with the rope, for example? (I mean you use two ropes, one is attached to the side of the electronics box and the other is attached to the end of the optical cell.) Of course, we need to confirm that the vibration status does not change much, and that the optical cell is strong enough.)

Right, my fear is about the optical cell vibrations. We would rather have the air flow from top to bottom or bottom to top, than in the horizontal, crossing the cell. This could cause turbulances which may distord the spectrum... We are planning additional flights in 2024, comparing with toher instruments. We hope that for those flights, we will be able to request a monger flight chain (though not yet sure we'll be granted our wish): 35 m instead of 17m. Then, we could see if there is an improvement or not, whcih would mean that the electronic could be a source (I suspect minor compared to the balloon at the descent).

---

## Author Response (AR2)

**Response to Proceeding review file validation:**

Figure 5 has been obtained plotting balloons trajectories in Google map. Following your suggestions, I have updated the figure such as:

[Figure]

The Copyright statement appears on the map.
For the Color Blindness Simulator: One color (orange) is dedicated to FPH trajectories and black is dedicated to Pico trajectories. I selected different shapes for different days.

Figure 11 has been checked by Color Blindness Simulator.